

# Validation of ACE-FTS version 5.2 ozone data with ozonesonde measurements

Jiansheng Zou[1], Kaley A. Walker[1], Patrick E. Sheese[1], Chris D. Boone[2], Ryan M. Stauffer[3], Anne M. Thompson[3], David W. Tarasick[4]

[1]Department of Physics, University of Toronto, Toronto, Ontario, Canada
       [2]Department of Chemistry, University of Waterloo, Waterloo, Ontario, Canada
       [3]NASA/Goddard Space Flight Center (GSFC), Greenbelt, MD, USA
       [4]Environment and Climate Change Canada, Downsview, Toronto, Ontario, Canada

*Correspondence to*: Kaley A. Walker (kaley.walker@utoronto.ca)

**Abstract.** Two decades of ACE-FTS version 5.2 (v5.2) ozone data (2004-2023) are evaluated with ozonesonde data from across the globe. The biases between the ACE-FTS and ozonesonde measurements are first estimated by analysing coincident data pairs. A second approach is taken for the validation by comparing the ACE-FTS and ozonesonde monthly mean time series, with the former generated by sampling the ACE-FTS data within latitude/longitude boxes (i.e., ±5°/±30°) surrounding the stations and calculating the monthly averages. The biases, correlations, variation patterns and the mean states of the two time series are compared. The biases estimated in this way exhibit more consistent and smoother features than using the coincident pair method. The ACE-FTS and ozonesonde monthly mean time series are highly correlated and exhibit similar variation patterns in the lower stratosphere at all latitudes. The ACE-FTS instrument drifts for each station are assessed in terms of the long-term linear trends relative to ozonesondes, which, although highly stable, may have their own minor changes with time.

The ACE-FTS ozone profiles exhibit in general high biases in the stratosphere, increasing with altitude up to ~10% at around 30 km, and have local maximum differences with ozonesonde profiles at the tropopause heights. The ACE-FTS instrument drifts are generally insignificant overall in the stratosphere with high variation between the stations. Averaging the individual station instrument drifts within several latitude bands results in small insignificant drifts of within ±1 % dec$^{-1}$ in the northern mid- to high latitudes, and the southern high latitudes, and a small positive

insignificant drift of 0 - 3 % dec$^{-1}$ in the tropics and southern mid-latitudes with overall uncertainties at 2 - 3 % dec$^{-1}$ (2σ level) in the low stratosphere. In the troposphere, the average ACE-FTS instrument drifts vary with altitude and exhibit large drifts between -10 and +10 % dec$^{-1}$ with uncertainties of 10 % dec$^{-1}$.

## 1 Introduction

ACE-FTS, the Atmospheric Chemistry Experiment - Fourier Transform Spectrometer, launched aboard the
Canadian satellite SciSat-1 on 13 August 2003, has measured atmospheric constituents for 20 years (e.g., Bernath et al., 2005). The ACE measurement period (2004 – present) overlaps the period of ozone recovery following the successful implementation of the Montreal Protocol in the late 1980s and its subsequent amendments, which prohibit the use of halogen-containing ozone depleting substances (ODSs) on the global scale (Salawitch et al., 2019), preventing further ozone depletion in the atmosphere. Observational evidence shows that the declining trends in ozone



stopped around 1997 and since 2000 ozone increases have been observed in the upper stratosphere and in total column ozone over the Antarctic as well (Salawitch et al., 2019; Steinbrecht et al., 2018). The reversal of the ozone trend is, however, not hemispherically symmetric and the ozone recovery rate is three to four times slower than the rate of decline prior to the 1990s (Steinbrecht et al., 2018). In the lower stratosphere, ozone has exhibited declining trends between 60°S and 60°N (Ball et al., 2017). The mechanisms causing these ozone changes can be complex. Besides

the reduction in releasing anthropogenic ODSs, climate change (Ball et al., 2020) plays a compounding role in that the increase of greenhouse gases (GHGs) causes cooling of the stratosphere, thus slowing the temperature-dependent ozone destruction processes and causing ozone increases in the upper stratosphere, while the increase of GHGs causes acceleration of the Brewer-Dobson Circulation (BDC), bringing more low ozone-concentration air from the troposphere to the low stratosphere. The negative ozone trends in the lower stratosphere from 60°S to 60°N, although

appearing as insignificant, were further verified by Godin-Beekmann et al. (2022) using multivariable linear regression analyses of multiple independent merged satellite datasets. Using several merged satellite ozone datasets, Szelag et al. (2020) investigated the seasonality of the ozone profile trends between 60°S and 60°N, confirming the positive trends in the upper stratosphere and revealing the strong variability of the sign of the ozone trend between positive and negative as a function of altitude and season in the equatorial region. These studies (e.g., Ball et al., 2017; Szelag et

al., 2020; Godin-Beekmann et al., 2022) are based on merged satellite data sets spanning multiple decades. The ACE-FTS data are part of some of these merged satellite datasets, e.g., GOZCARDS (Froidvaux et al., 2015) and SAGE-CCI-OMPS (Sofieva et al., 2017). In contrast, in Thompson et al. (2021; update in Stauffer et al. 2024) tropical ozonesondes were found not to show a net annual trend in the lowermost stratosphere (100 - 50 hPa). Although there are 5-15 % dec[-1] decreases in ozone in this region during July-September over stations such as Samoa, Paramaribo,

Ascension and Kuala Lumpur, the tropopause height during this period has increased more than 100 m dec[-1]. When tropopause-height referenced ozone columns are analyzed, the trend in the lowermost stratosphere disappears. These different conclusions for ozone trends in the lowermost stratosphere can be drawn depending on how the tropopause height is referenced.

        A comprehensive validation study of ACE-FTS v4.1/4.2 ozone data in the stratosphere and mesosphere has

been carried out with satellite instrument datasets by Sheese et al. (2022). This current study compares the new ACE-FTS v5.2 ozone data with global ozonesonde data to assess the data quality in the upper troposphere and lower stratosphere (UTLS). For the UTLS region, ozonesonde records have been widely used as a reference for validating other instruments such as satellite instruments (e.g., Thompson et al., 2021). Ozonesondes have a long history of use for ozone measurements, for example, Resolute Bay ozone sounding began routine recording in 1966 and has the

longest record in the world (Tarasick et al., 2016). Despite its long history, the ozonesonde measurements have been experiencing hardware changes and processing corrections from time to time (Tarasick et al., 2016). Nevertheless, because of its reliability and its availability at low altitudes, some researchers (e.g., Adams et al., 2013; Bognar et al., 2019) have used ozonesonde data to extend satellite measurements such as ACE-FTS ozone profiles to the ground level to calculate the total column ozone (TCO) for ground-based comparisons. Ozonesonde data have often been used

to estimate the biases and long-term stability of limb-viewing satellite ozone datasets such as ACE-FTS v2.2 update (Dupuy et al., 2009), Aura-MLS (Microwave Limb Sounder) v2.2 (Jiang et al., 2007), MIPAS (Michelson



Interferometer for Passive Atmospheric Sounding) - Envisat IMK-IAA version V3O_O3_7 for the HR (High Resolution) period (July 2002- March 2004) (Steck et al., 2007), SCIAMACHY (SCanning Imaging Absorption SpectroMeter for Atmospheric CHartographY) - Envisat v2.9 and v3.0 (Jia et al., 2015), OMPS (Ozone Mapping Profiler Suite) Antarctic data from both the Limb Profiler (LP) version 1 and Nadir Profiler (NP) version 1 (Kramarova et al., 2014), and SAGE III/ISS v5.1 ozone profile data (McCormick et al., 2020; Wang et al., 2021) as well as a collection of 14 limb scatter, limb emission, and solar occultation datasets (Hubert et al., 2016). Note that for solar occultation instruments such as ACE-FTS and SAGE III/ISS, there are fewer profiles (two profiles per orbit) than for other limb viewing instruments, whose sampling densities are much higher, e.g., 240 profiles per orbit for Aura-MLS. When using coincident data pairs to compare satellite and ozonesonde data, there are no definitive criteria to select coincident data pairs. For limb emission or limb solar scattering instruments, it can be sufficient to use relatively tight temporal criteria such as time differences within ±6 hours, while for solar occultation instruments relatively relaxed criteria such as ±24 hours of time differences are often used (e.g., Dupuy et al., 2009; McCormick et al., 2020).

In recent years the data quality of ozonesonde measurements at some stations has been an issue. Stauffer et al. (2020) found that about 14 stations using electrochemical concentration cell (ECC) ozonesondes, primarily in Canada and the tropics, experienced TCO drop-offs after 2013. The drop-off was found to be associated with only one type of ECC device, that from the manufacturer Environmental Science (EnSci) after certain production lots (serial numbers) (Stauffer et al, 2022; Nakano and Morofuji, 2023). With the progress in understanding of the drop-off issue, Stauffer et al. (2022) showed that using recently updated data from 37 stations, which have been re-processed through the homogenization process, overall improvements in the long-term ozone data quality have been achieved, particularly for those at the Canadian sites. The ozonesonde network data continue to be a reliable data source with caveats only for a subset of drop-off stations in the tropics and subtropics (Stauffer et al., 2022).

In this study, the ACE-FTS version 5.2 ozone data spanning almost 20 years (2004-2023) are compared with the global ozonesonde data to derive the biases and drifts of ACE-FTS ozone measurements relative to the ozonesonde measurements. Section 2 describes the ACE-FTS ozone data products and the ozonesonde data from the four data centres: Network for the Detection of Atmospheric Composition Change (NDACC), World Ozone and Ultraviolet Radiation Data Centre (WOUDC), Southern Hemisphere ADditional OZonesondes (SHADOZ), and Harmonization and Evaluation of Ground-based Instruments for Free Tropospheric Ozone Measurements (HEGIFTOM). Section 3 introduces the method of selecting coincident data pairs, the method of generating ACE-FTS time series around an ozonesonde station, the formulae for calculating the biases and errors, the linear trends and standard errors, and the formulae used for the aggregated averages. In Sect. 3, a summary of the drop-off analysis based on the ACE-FTS data is also given. This is the basis used to select stations for the data analyses presented in Sect. 4, including bias determination, time series comparisons, and drift determination. The conclusions are given in Sect. 5. In Appendix A and supplement details for the drop-off analysis are given.



## 2 ACE-FTS and ozonesonde data sets

### 2.1 ACE-FTS ozone data products

ACE-FTS is an infrared solar occultation instrument onboard the Canadian satellite Scisat-1 which follows a non-sun synchronous orbit with high inclination (Bernath et al., 2005). ACE circles the Earth approximately 15 times per day and takes 2 profile measurements per orbit with the highest density of samples towards high latitudes. This study uses the most recent ozone profile data from the ACE-FTS v5.2 (Boone et al., 2023), which evolved from the previous versions, e.g., v2.2 (Boone et al., 2005), v3.0 through v3.5/3.6 (Boone et al., 2013), and v4.1/4.2 (Boone et al., 2020).

Initial ACE-FTS v1.0 ozone data was first validated with other satellite instruments (SAGE III/Meteor-3M and POAM III) (Walker et al., 2005). The ACE-FTS version v2.2 ozone update product was validated with nearly 20 satellite-, balloon-, air-borne and ground-based instruments including global ozonesondes to determine the ACE-FTS biases against other instruments for the period 2004-2009 (Dupuy et al., 2009). In Sheese et al. (2017), ACE-FTS v3.5 data for $O_3$, $N_2O$, $H_2O$, $HNO_3$ and CO were compared with collocated data from MIPAS and Aura-MLS for the period February 2004 to April 2012 and the comparisons show that ACE ozone is within ±5% of MIPAS and Aura-MLS data in the mid-stratosphere and exhibits a positive bias of 10-20% in the upper stratosphere. Sheese et al. (2022) extended the ACE-FTS ozone data validation study with 5 satellite datasets for both v3.5/3.6 and v4.1/4.2 to assess biases and drifts and concluded that v4.1/4.2 showed slightly larger biases than v3.5/3.6 in the middle to upper stratosphere, but is stable to within 1% $dec^{-1}$, whereas v3.5/3.6 exhibited a significant negative drift on the order of 1-3% $dec^{-1}$.

This work utilizes the ACE-FTS v5.2 ozone volume mixing ratio (VMR) profile data provided on a 1-km regular grid from 0.5 to 149.5 km with the actual retrieval extending from 5 to 95 km. An additional flag file, in which the data quality of each profile point is assigned a flag value from 0 to 9 based on Sheese et al. (2015), was used to filter the data. The ACE-FTS profile data with flag values greater than 2 were removed such that bad profiles with excessive retrieval statistical fitting error, physically unrealistic outliers and known instrumental/processing errors were discarded (Sheese et al., 2015).

### 2.2 Ozonesonde data

An ozonesonde is a small, lightweight, balloon-borne instrument consisting of a pump and an ozone sensing cell coupled to a standard meteorological radiosonde to measure ozone concentration and ambient air pressure, temperature, humidity, and other meteorological parameters. The measurement principle is based on the redox reaction of ozone with potassium iodide in aqueous solution, which releases free iodine molecules $I_2$ (e.g., Tarasick et al., 2016). With the free $I_2$ molecules in the sensing cell and the anode and cathode at the two ends, an electric current, which is proportional to the ozone molecule amount, is generated and this measurement can be converted to the ozone amount (Tarasick et al, 2016). Different ozonesonde instrumentation has been used in the networks and over time. In the case of ozonesondes at Canadian sites, Brewer-Mast-type ozonesondes were predominant before the 1980s, thereafter ECC ozonesondes have been used. The ozonesonde data used in this study are all ECC-type except at





Hohepeissenberg where Brewer-Mast ozonesondes are used. The ozonesonde measures ozone concentrations from
the ground up to ~32 km before the balloon bursts. The vertical intervals of the data records may vary from a few
meters to hundreds of meters. However, these data are not necessarily independent, and the vertical resolution is
dependent on the response time of the sonde and the rise rate of the balloon and is therefore about 100 – 150 m.

The most recent quality-assured ozonesonde data reported from four data centers, NDACC, WOUDC,
SHADOZ, and HEGIFTOM, are used in this validation study. The locations of those ozonesonde stations are listed
in Fig. 1 and Table 1. The SHADOZ ozonesonde data (Witte et al., 2017, Thompson et al., 2017, Witte et al. 2018,
Sterling et al., 2017), managed by NASA GSFC, are composed of a network of ozonesonde stations operating in the
tropics, subtropics, and the southern hemisphere. The ozoneonde data in the tropical area are all taken from SHADOZ.
The extra-tropical sondes are taken from NDACC and WOUDC, with the former managed by a Steering Committee
under which various working groups are responsible for the access and standardization of the data, and the latter
operated by the Meteorological Service of Canada (MSC), a branch of Environment and Climate Change Canada
(ECCC). For those stations having both NDACC and WOUDC datasets, only the dataset with the larger volume is
used (see Table 1). For the Canadian sites, the data downloaded from HEGIFTOM, which are the products after the
recent homogenization, are finally used in the analysis.

In the analysis of ozonesonde records, the first step is to reject bad data points such as negative or excessively
large values of ozone partial pressure (typically occurring at only a few stations), and to unify the units of temperature,
pressure, geo-potential height, altitude, and ozone partial pressure. In this study, there is no limit on the data altitude,
e.g., including data points above 10 hPa (32 km) as well as data profiles with top altitudes below 30 hPa (25 km), thus,
allowing more data to be used at lower altitudes while continuing examination of profiles at high altitudes, e.g., > 32
km. In the results the data points at top altitudes sometimes appear to be outliers, and it is due to the limit at high
altitude measurements before the balloon bursts. However, these top altitude thresholds are not fixed. The next step is
to apply a low pass filter to get smoothed ozonesonde profiles at the ACE-FTS regular 1-km grid points. A Gaussian
function $\exp(-\Delta z^2 / 2s^2) / \sqrt{2\pi s}$ was applied to the ozonesonde profile data, where $\Delta z$ is the distance between an
ozonesonde data point and an ACE-FTS point on the regular 1-km grid, $s$ set to be 1 km, which makes a data point at
1-km distance have a weighting coefficient to be 0.66. Other similar but slightly different approaches for the smoothing
(convolution) of the high vertical resolution ozonesonde data have been used in other studies (e.g., Dupuy et al., 2009;
Kar et al., 2007). Thus, the smoothed ozonesonde profile data have the same grid as the ACE-FTS, ready for the
calculations in the next sections.



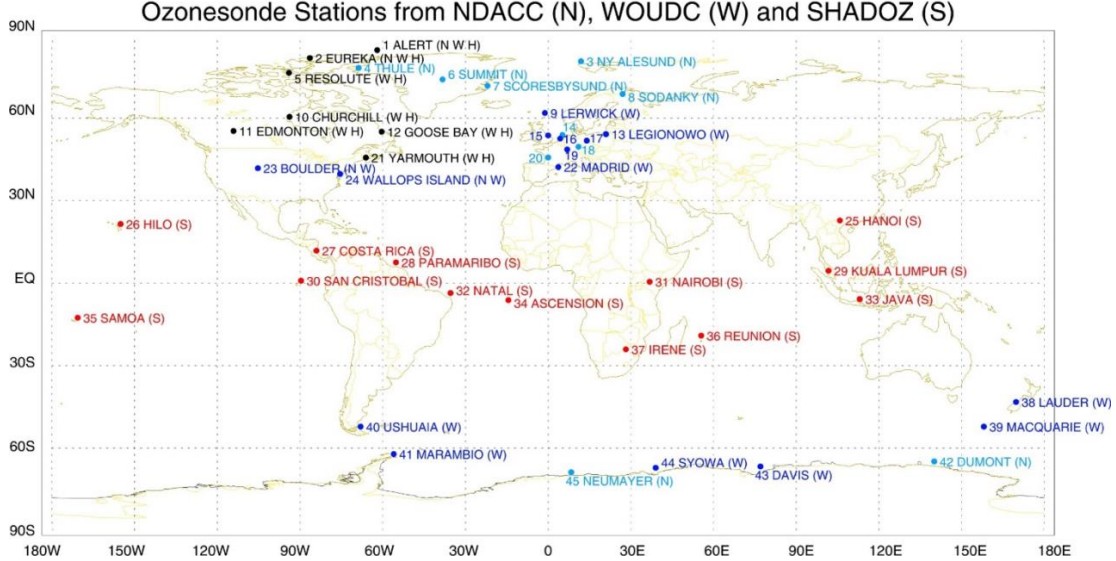

Figure 1 The ozonesonde stations from the four networks: NDACC (N), WOUDC (W), SHADOZ (S) and HEGIFTOM (H). At some stations multiple data sources are available with the last letter denoting the data source used in the analysis. The stations from NDACC (N) are written in light blue, from WOUDC (W) in dark blue, from SHADOZ (S) in red, and from HEGIFTOM (H) in black. The stations 14-20 are all located in Europe and their names are listed at the top of the figure.



| Station | Location | Source | # of profiles | Ncoin | Nmon | Drop-off Flag |
|---|---|---|---|---|---|---|
| Alert | 82.5°N, 62.3°W | N,W,H | 856 | 103 | 77 | 2 |
| Eureka | 80.1°N, 86.4°W | N,W,H | 1194 | 554 | 77 | 2 |
| Ny Ålesund | 78.9°N, 11.9°E | N | 1452 | 214 | 74 | 3 |
| Thule | 76.5°N, 68.7°W | N | 128 | 28 | 18 | 9 |
| Resolute | 74.7°N, 95.0°W | W,H | 596 | 81 | 79 | 2 |
| Summit | 72.6°N, 38.5°W | N | 634 | 76 | 85 | 2 |
| Scoresbysund | 70.5°N,22.0°W | N | 968 | 136 | 148 | 1 |
| Sodankylä | 67.37°N, 26.67°W | N | 694 | 128 | 124 | 2 |
| Lerwick | 60.1°N, 1.18°W | W | 769 | 96 | 120 | 3 |
| Churchill | 58.8°N, 94.7°W | W,H | 528 | 39 | 151 | 1 |
| Edmonton | 53.6°N, 114.1°W | W,H | 882 | 56 | 185 | 2 |
| Goose Bay | 53.3°N, 60.4°E | W,H | 836 | 60 | 171 | 2 |
| Legionowo | 52.4°N, 21.0°W | W | 916 | 42 | 165 | 3 |
| De Bilt | 52.1°N,5.1°E | W,N | 924 | 39 | 166 | 3 |
| Valentia | 51.93°N, 0°E | W | 546 | 28 | 145 | 3 |
| Uccle | 50.8°N, 4.4°E | N,W | 3060 | 143 | 175 | 2 |
| Prague | 50.0°N, 14.5°E | N,W | 958 | 22 | 61 | 3 |
| Hohenpeißenberg | 47.8°N, 11.0°E | N | 2426 | 115 | 174 | 3 |
| Payerne | 46.5°N, 6.6°E | W | 3011 | 117 | 170 | 3 |
| Haute-Provence (OHP) | 43.94°N, 5.71°E | N | 291 | 7 | 71 | 2 |
| Yarmouth | 43.9°N, 66.1°W | W,H | 848 | 43 | 167 | 1 |
| Madrid | 40.8°N, 12.2°W | W | 940 | 38 | 166 | 3 |
| Boulder | 40.03°N, 105.25°W | N,W | 802 | 25 | 117 | 2 |
| Wallops Island | 37.9°N, 75.5°W | N,W | 731 | 27 | 104 | 3 |
| Hanoi | 21.0°N, 105.8°E | S | 337 | 6 | 77 | 1 |
| Hilo | 19.7°N, 155.07°W | S | 879 | 15 | 84 | 2 |
| Costa Rica | 10°N, 84°W | S | 623 | 7 | 64 | 1 |
| Paramaribo | 5.8°N, 55.2°W | S | 605 | 8 | 62 | 3 |
| Kuala Lumpur | 2.7°N, 101.7°E | S | 332 | 0 | 55 | 2 |
| San Cristobal | 0.9°S, 89.6°W | S | 215 | 0 | 24 | 9 |
| Nairobi | 1.3°S, 36.8°E | S | 679 | 7 | 57 | 2 |
| Natal | 5.4°S, 35.4°W | S | 496 | 9 | 53 | 3 |
| Java | 7.6°S, 113°E | S | 175 | 0 | 38 | 9 |





| Ascension | 8.0°S, 14.4°W | S | 552 | 4 | 50 | 1 |
|---|---|---|---|---|---|---|
| Samoa | 14.3°S, 170.6°W | S | 592 | 14 | 71 | 2 |
| Reunion | 20.9°S, 55.5°E | S | 516 | 8 | 82 | 2 |
| Irene | 25.9°S, 28.2°E | S | 259 | 8 | 59 | 3 |
| Lauder | 45.0°S, 169.7°E | W | 948 | 37 | 163 | 2 |
| Macquarie | 54°S, 158°E | W | 960 | 53 | 190 | 2 |
| Ushuaia | 54°S, 68°W | W | 223 | 10 | 75 | 3 |
| Marambio | 64.2°S, 56.7°W | W | 908 | 136 | 135 | 2 |
| Dumont | 66.67°S, 140.01°E | N | 375 | 75 | 116 | 3 |
| Davis | 68.6°S, 78.0°E | W | 647 | 134 | 133 | 3 |
| Syowa | 69°S, 39.6°E | W | 653 | 121 | 104 | 2 |
| Neumayer | 70.7°S, 8.3°W | W | 1320 | 128 | 149 | 3 |

**Table 1: The ozonesonde stations from the NDACC (N), WOUDC (W), SHADOZ (S), and HEGIFTOM (H) networks indicated by the letter in the parenthesis. For some stations with multiple sources, the last letters in column 3 indicate the data sources used in the study. Column 4 denotes the numbers of ozonesonde measurements during 2004 – 2023, column 5 the numbers of coincident data pairs, column 6 the numbers of common months in the ozonesonde and ACE-FTS monthly mean time series. Column 7 lists the drop-off flags: 1 for drop-off and 2 for non-drop-off stations as identified in Appendix A, 3 for not subject to drop-off analysis and deemed as non-drop-off stations, 9 having issues of data gaps in analysis.**

## 3 Methods

### 3.1 Determining the biases using coincident data

The first validation task for this work is to determine the biases of the ACE-FTS ozone profiles versus coincident ozonesonde measurements. As often applied for satellite validations (e.g., Jiang et al., 2007; Dupuy et al., 2009), the method of using coincident data, i.e., sets of data pairs from the two instruments at close times and locations, to calculate the ensemble mean differences, is used in this study. As there are no perfectly coincident pairs (Sheese et al., 2020), the differences calculated from these coincident pairs actually consist of contributions from the random and systematic errors of the two instruments, the sampling error due to the mismatch in time and space of the coincident pair, which is also called the "geophysical variability", or the "natural variability", and the smoothing error due to the different footprints of the measurements (Laeng et al., 2021). Geophysical variability is impacted by the choice of coincidence criteria for the data pairs. Intuitively, the smaller the temporal and spatial separations are, the lower the geophysical variability is, but on the other hand, fewer sampled data are used. A balance is required whether setting stricter or more relaxed criteria. Sheese et al. (2020) showed that the ozone geophysical variability is independent of the chosen time criterion up to 12 hours in the lower stratosphere, and conversely, in the upper stratosphere the geophysical variability tends to be independent of the chosen distance separation up to within 2000 km. In a case study, Laeng et al. (2021) showed that the geophysical variability grows with time difference for values smaller than 5 hours but becomes stabilized for time differences larger than 10 hours; and as for the spatial differences, the



geophysical variability increases with the spatial distance and becomes exceptionally large as spatial distance exceeds a certain threshold such as 1000 km. So, the spatial difference should be limited. Although these results are derived from high-resolution models, they provide some insight into choosing coincident pairs.

As a solar occultation sounder, ACE has sparse sampling in time and space (~15 sunrise and 15 sunset occultations per day), and due to the ACE orbit the measurements are densest towards high latitudes and sample less frequently at low latitudes. If coincidence criteria are set too tight, there will be few coincident data pairs for many stations at lower latitudes. In this study, the upper limits are set for time differences of within ±24 hours, and for latitude/longitude ranges of ±5°/±15° with adjustments depending on the latitude (see Sect. 4.1). After collecting the coincident data pairs, each individual difference $\delta_i(z) = ACE_i(z) - Sonde_i(z), i = 1, ..., N(z),$ is calculated,

here $z$ is the altitude, $N(z)$ the number of coincident pairs at $z$, and the pair ($ACE_i(z)$, $Sonde_i(z)$) is the $i$-th coincident ACE-FTS and ozonesonde measurements, e.g., VMR values, at $z$. The mean absolute difference profile is obtained by $\bar{\delta}(z) = \sum_{i=1}^{N(z)} \delta_i(z) / N(z)$. For simplicity since the variables are always a function of $z$, the argument $z$ is omitted in the following formulae. The mean relative difference profile is thus given by $\bar{\delta}^{rel} = \sum_{i=1}^{N} \delta_i^{rel} / N$, where the individual relative difference is calculated as $\delta_i^{rel} = \dfrac{\delta_i}{\sum_{j=1}^{N} (ACE_j + Sonde_j)/(2N)}$, and the denominator is the

ensemble mean of the ACE-FTS and ozonesonde values to suppress possible impacts from some outliers (Sheese et al, 2022). Letting $\sigma$ and $\sigma^{rel}$ be the de-biased standard deviations of $\delta_i$ and $\delta_i^{rel}$ time series, $\sigma = \sqrt{\sum_{i=1}^{N} (\delta_i - \bar{\delta})^2 / (N-1)}$ and, in a similar way, we calculate $\sigma^{rel}$. The standard errors of the estimated means $\bar{\delta}$ and $\bar{\delta}^{rel}$ are $s_{\bar{\delta}} = \sigma / \sqrt{N}$ and $s_{\bar{\delta}}^{rel} = \sigma^{rel} / \sqrt{N}$.

**3.2 ACE-FTS and ozonesonde monthly mean time series**

Besides using "instantaneous" coincident pairs, the monthly mean comparison method is utilized for bias assessment in this study. For an ozonesonde station, the monthly mean-time series can be calculated from its records. Typically, the frequency of ozonesonde launches is about once per week for most stations. For the comparison, the ACE-FTS data points within certain latitude and longitude ranges surrounding each of the ozonesonde stations are used to generate the monthly average time series.

In choosing the latitude/longitude range with the ozonesonde station at its center, the widths of the boxes were examined to test the results. The latitude width was chosen ±5° of the ozonesonde station latitude. The various values were tried for the longitude widths, ±10°, ±20°, ±30°, and ±45°. Because of the generally zonal distribution of atmospheric species, the ACE-FTS time series generated using these longitude ranges capture roughly similar temporal variations of time scales of several months with differences in detailed features at smaller time scales. For





comparison with ozonesonde records that will be averaged monthly, the longitude range of ±30° was selected for this study as a trade-off between collocating with the ozonesonde station and obtaining sufficient data points. Figures 2a and 2b are examples of monthly mean time series at Nairobi and Hohenpeißenberg, representative of low and mid-latitude stations, showing the ACE-FTS and ozonesonde records at 23.5 km. Nairobi is located near the equator where ACE has fewer samples than at higher latitudes, such as near Hohenpeißenberg. Also, 23.5 km is an altitude where

active atmospheric dynamic motions occur. The Nairobi time series is characterized by the equatorial quasi-biennial oscillation (QBO), and Hohenpeißenberg time series shows the annual cycle strongly. Both the ACE-FTS and ozonesonde time series agree very well at these two stations. The absolute and relative differences between the two-time series can be derived using the same formulae outlined in Sect. 3.1 by treating the same months as coincidences.

### 3.3 The linear trend estimation

During the ACE mission (2004 – present), ozone is in its recovery phase (e.g., Steinbrecht et al. 2018). Estimating the ACE-FTS instrument long-term drifts relative to the ozonesonde measurements is another objective of this study. A linear model is chosen for the instrument drift. This is in line with the recommendation of the LOTUS (Long-term Ozone Trends and Uncertainties in the Stratosphere) regression model, in which long-term ozone trends after 2000 are fitted as linear trends (SPARC/IO3C/GAW, 2019). As a result, the instrument drifts are also fitted with

linear trends. The ozone time series contain not only long-term trends, but also other temporal variation components such as annual and semi-annual cycles, the QBO, El Niño Southern Oscillation (ENSO), etc. One can separate the linear trends from the cyclic components by employing multivariable linear regression to the ozone time series (Eckert et al., 2014; Moreira et al., 2016; Toihir et al., 2018). In the ACE-FTS time series (as generated in this study), there are often data gaps in certain months and the gaps occur yearly throughout the entire mission, because the ACE orbit

essentially repeats every year. Therefore, there may not be enough data points to emulate the cyclic components at some latitudes. Since the annual cycle is the largest temporal component outside of the equatorial region, de-seasonalized time series are used to calculate the linear trends (SPARC/IO3C/GAW, 2019). An annual cycle was generated by averaging the data for each month over the entire period, which allows data gaps in certain months, and then removed from the monthly mean time series, resulting in de-seasonalized monthly mean time series. Fig. 3a and

3b are examples of de-seasonalized time series for Nairobi and Hohenpeißenberg at 23.5 km derived from the data in Fig. 2a and 2b, respectively. At Nairobi, the QBO features are still dominant, while at Hohenpeißenberg, the annual cycles are largely removed, leaving residual signals of other temporal components with smaller amplitudes. As the time series are nearly two decades long, the shorter-term variations may have small impact on the long-term trend estimation. For illustration, the lines shown in Fig. 3a and 3b are the linear trends obtained by fitting the differences

between the ACE-FTS and ozonesonde de-seasonalized time series, i.e., the ACE-FTS instrument drift relative to the ozonesonde (herein called the instrument drift). The calculation for the instrument drift is based on the formulae below.

a)



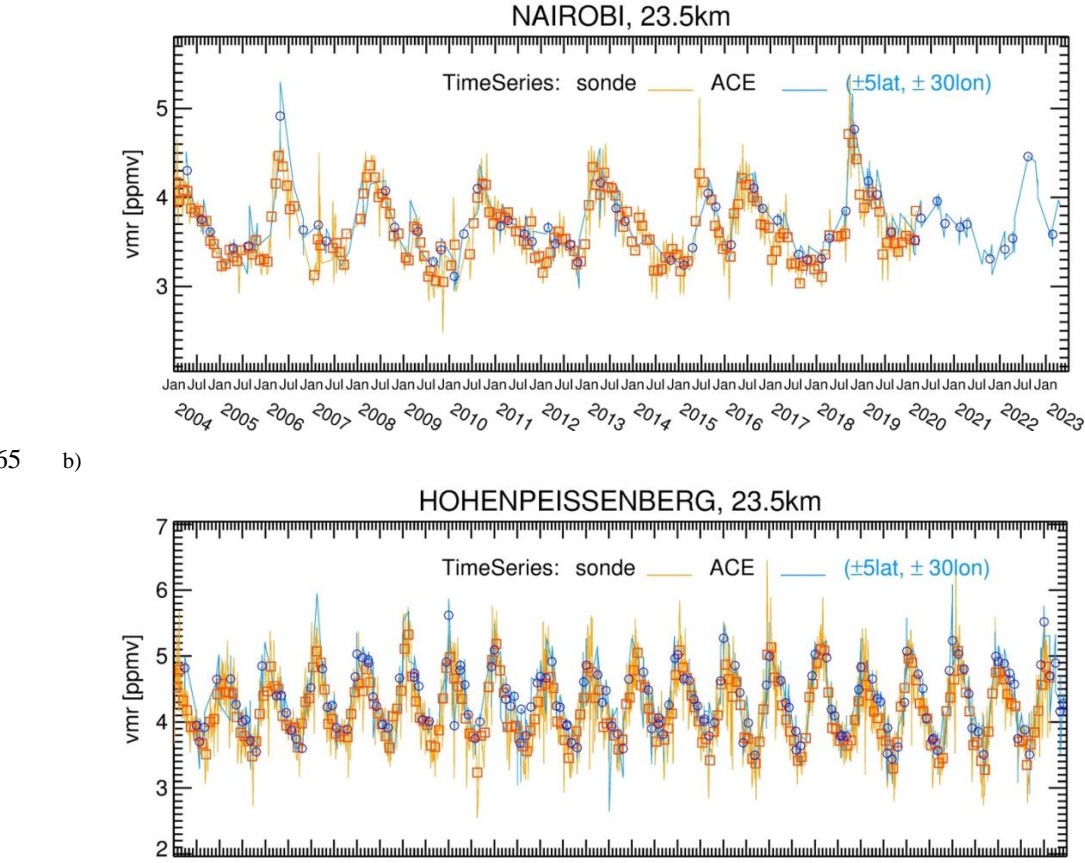

b)

**Figure 2: The ACE-FTS ozone VMR time series (blue curves) from the data points at 23.5 km collected over the areas of the latitude/longitude range of ±5°/±30° around Nairobi (1.3°S, 36.8°E) (a) and Hohenpeißenberg (47.8°N, 11.0°E) (b) together with the monthly means (blue circles). Shown also are the corresponding ozonesonde records (yellow curves) with**
**the monthly means (red squares).**

Letting $y_i = a + bt_i + \varepsilon_i$, $i=1\ldots, N$ represent a time series with time $t_i$ at month $i$, with $y_i$ the differences between the two de-seasonalized monthly mean time series, $a$ the initial difference term, and $b$ the instrument drift. The simple least squares regression gives estimate of $b$ and $a$, $\hat{b} = \sum_{i=1}^{N} y_i (t_i - \bar{t}) / \sum_{i=1}^{N} (t_i - \bar{t})^2$ and $\hat{a} = \bar{y} - \hat{b}\bar{t}$, where $\bar{y}$ and $\bar{t}$ are the means of $y_i$ and $t_i$. The residual errors are $\varepsilon_i = y_i - \hat{a} - \hat{b} t_i$ and the standard error of $\hat{b}$ is

$s_{\hat{b}} = \sqrt{\sum_{i=1}^{N} \varepsilon_i^2 / (N-2) / \sum_{i=i}^{N} (t_i - \bar{t})^2}$ with $\hat{b} \pm 2s_{\hat{b}}$ representing the 95% confidence level for $\hat{b}$.



a)

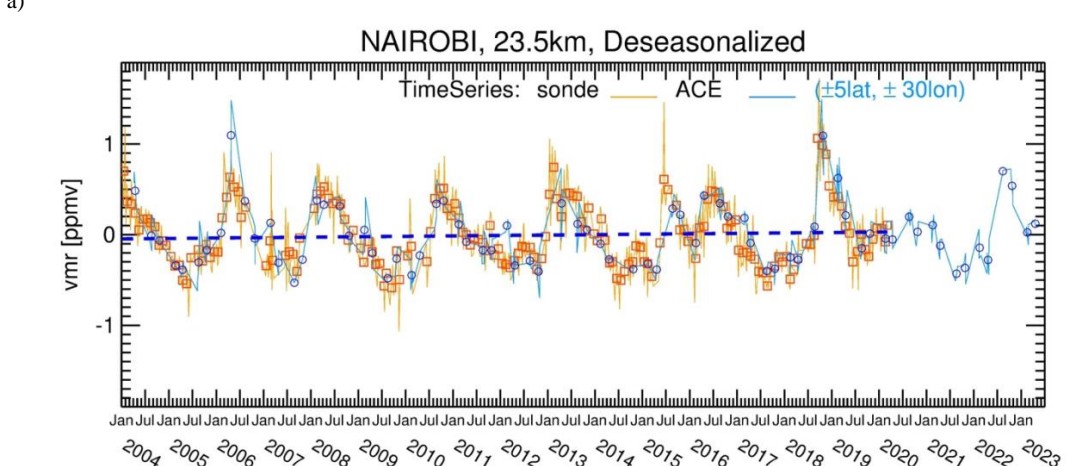

b)

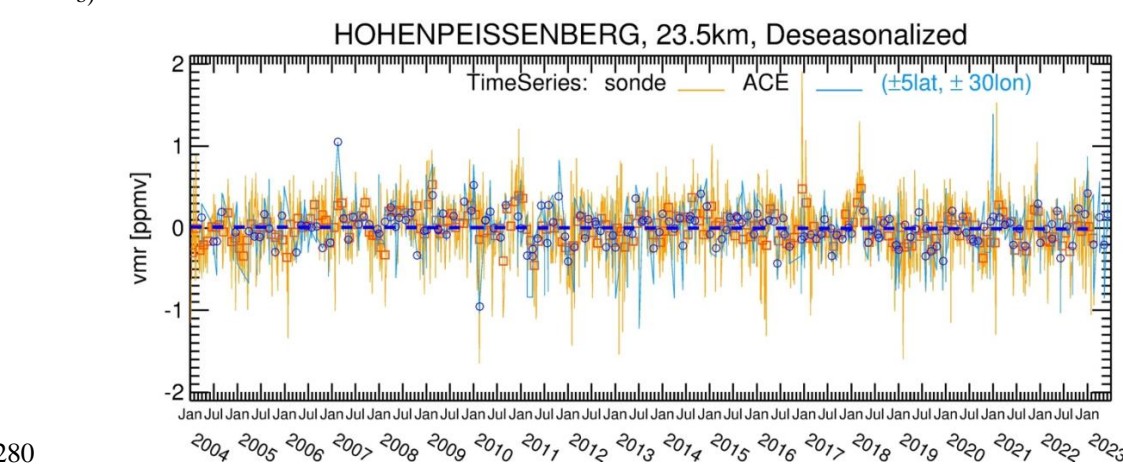


**Figure 3: The de-seasonalized ACE-FTS (blue curves) and ozonesonde (yellow curves) ozone VMR time series at 23.5 km at Nairobi (a, top) and Hohenpeissenberg (b, bottom) and their respective monthly means (blue circles and red squares). The dark blue dashed lines are the linear fits to the differences of ACE-FTS and ozonesonde de-seasonalized monthly mean time series, defined as the instrument drift.**

**3.4 Averaging over latitude bands**

To highlight the validation results from more than 40 stations, individual comparisons are grouped into several latitude band averages. For each latitude band, the aggregated statistical quantities can be expressed in terms of the individual station quantities $\overline{\delta}_k, \sigma_k, N_k$, i.e., the mean, standard deviation, and the number of measurement points at station $k$, respectively. The aggregated mean can then be expressed as the weighted average over the





individual means, $\tilde{\delta} = \sum_k \bar{\delta}_k w_k / \sum_k w_k$. There are different ways to choose $w_k$, such as $N_k$ or the inverse-variance

of the sample data $1/\sigma_k^2$ (Sheese et al., 2022). This study uses the first approach, which is equivalent to averaging

the entire data set for each latitude band with all points equally weighted and the resulting statistical formulae are

straight       forward.     The      standard      deviation      of      the      aggregated      data      set      is

$\tilde{\sigma} = \sqrt{\sum_k [\sigma_k^2 (N_k - 1) + N_k (\tilde{\delta} - \bar{\delta}_k)^2] / (\sum_k N_k - 1)}$ and the standard error of the aggregated mean $\tilde{\delta}$ is

$\tilde{s}_{\tilde{\delta}} = \tilde{\sigma} / \sqrt{\sum_k N_k}$. For the calculation of relative differences, the formulae follow from the above equations.

To obtain the aggregated mean of the instrument drifts from the individual stations within a latitude band, a

formula for a scalar weighted average (Hubert et al., 2016; Sheese et al., 2022) is adopted, $\tilde{b} = \sum_k w_k \hat{b}_k / \sum_k w_k$

, where the weighting factor $w_k = 1 / s_{\hat{b},k}^2$, $s_{\hat{b},k}$ is the estimated standard error of the linear drift $\hat{b}_k$ at station $k$ (Sect.

3.3) and the aggregated standard error for $\tilde{b}$ is $\tilde{s}_{\tilde{b}} = 1 / \sqrt{\sum_k w_k}$.

### 3.5 Ozonesonde data selection for trend calculations


Among the ozonesonde stations, only those ozonesonde records overlapping with the ACE period (2004 -
2023) and having a time span longer than 10 years between 2004 and 2023 are selected for the drift analysis. Both
ACE-FTS and ozonesonde monthly mean time series are calculated in the months common to both datasets.

### 3.6 Selection of "drop-off" ozonesonde stations

The current study uses the ozonesonde data as reference for ACE-FTS data validation. It is also interesting
to use ACE-FTS data to examine the drop-off features from the perspective of employing solar occultation
measurement data. Detailed drop-off analyses are given in the Appendix and the supplement. The results are
summarized in Table 1 with flags in column 7 highlighting the drop-off status verified with ACE-FTS data. Flag 1
indicates a likely drop-off station (with drop-off magnitude 3 – 4 %). They are operated with EnSci sondes and

comprise some Canadian sites, Churchill, and Yarmouth, Greenland's Scoresbysund, and the tropical stations at
Hanoi, Costa Rica, and Ascension.  The result (Table 1 and Table A1 in the Appendix) is generally consistent with
Stauffer et al. (2022) with differences for some tropical stations. Stauffer et al. (2022) found Hanoi is not a drop-off
site, while Samoa is, and Ascension is denoted by N/A due to insufficient comparison points. In an earlier analysis,
Hilo was found to be a drop-off station (Thompson et al., 2021). Despite a few different drop-off identifications, the

drop-off sites determined using ACE-FTS data are adopted in the following analyses for consistency. Flag 2 indicates
those stations operated with EnSci ECC sondes, but not likely affected with drop-off features. The stations with flag
3 are operated mostly with ECC sondes manufactured by Science Pump Corporation (SPC), which are unlikely
affected by drop-offs (Stauffer et al., 2022). Stations with flag 9, Thule, San Cristobal, and Java, have issues with
missing data which have impacts on trend and drop-off analyses, but are valid for bias analyses (see Sect. 4.2).



**3.7 The polar spring data**

To evaluate the impact of data within the polar vortices on these validation results such as on the instrument drift estimation, different tests were performed by including and excluding data within the polar vortices. In this study, a coarse definition was used to define periods affected by the polar vortex such that the data from January to March, north of 65°N and from September to November, south of 65°S, are regarded as within the polar vortices. Test results obtained by including or excluding the polar vortex data points do not show significant differences. For the results reported below, no additional data screening was made to exclude data within the polar vortex.

**4 Results**

**4.1 Biases from coincident data pair comparisons**

Given the background of the geophysical variability as discussed above, this paper presents results using a coincidence criterion that is latitude-dependent for the distance separation and time differences that are within ±24 hours for all latitudes: at 75°-90°N (S), spatial distances between ACE-FTS measurements and ozonesonde stations are < 500 km; at 60°-75°N (S), spatial distances are < 800 km; and at other latitudes, the latitude and longitude differences are ±5° and ±10°, respectively. This criterion is generally comparable to other studies. Dupuy et al. (2009) adopted a criterion of ±24 hours and 800 km as coincidence criteria to validate ACE-FTS v2.2 ozone update data using ozonesondes. McCormick et al. (2020) used ±24 hours in time, ±5° in latitude and ±10° in longitude for validating ozone data from the SAGE III/ISS solar occultation instrument, using the ozonesonde data from Lauder and Hohenpeißenberg. When validating satellite data with higher sampling density, the criteria can be stricter such as those used in Sepúlveda et al. (2021), where the coincidence criteria were chosen as ±12 hours and 500 km when comparing OMPS-LP ozone data with ozonesonde measurements at Antarctic stations, because OMPS measures 160-180 profiles in each orbit.



**Figure 4** The mean difference profiles between ACE-FTS and ozonesondes calculated from the coincident data pairs of ozone VMR vertical profiles in ppmv (ACE – ozonesonde). The dotted profiles are those from drop-off stations. The panels in the 2nd and 4th rows are expanded views of the lower altitudes (below 20 km) of the panels in the 1st and 3rd rows, respectively. The ozonesonde stations are grouped in latitude ranges as indicated on the top of set of panels. Different ozonesonde stations are indicated by different colors. The numbers of coincident pairs in the mean difference profiles are



**indicated at the left side at a 3 km interval using the same color codes as for the stations. The horizontal bars plotted every 3 km are ±2 $s_{\bar{\delta}}$ , $s_{\bar{\delta}}$ the standard errors of the estimated means. The horizontal dashed blue lines are the average tropopause heights in the latitude bands estimated from the temperature profiles in the ozonesonde data.**

350   Figures 4a - 4j show the ACE-FTS and ozonesonde mean difference profiles in ppmv, together with the standard error bars $\pm 2s_{\bar{\delta}}$ calculated from the coincident data pairs at the individual stations identified in Fig. 1. It should be noted that at six stations, Scoresbysund (Fig. 4b), Churchill (Fig. 4c), Yarmouth (Fig. 4e), Hanoi (Fig. 4f), Costa Rica (Fig. 4f) and Asension (Fig. 4h), the ozonesonde measurements have some low values in some periods after 2013 (e.g., likely impacted by drop-offs), and the differences at these stations are plotted with dotted lines. Each

figure contains two panels with the lower one replotting the lower altitude sections (<20 km) with an enlarged x-axis scale. In all the plots, the tropopause heights divide the stratospheric and tropospheric regions, which are calculated using ozonesonde temperature profile data according to the WMO 1957 definition as the lowest level at which the lapse rate decreases to 2°C/km or less. The stations, starting from the northernmost one in latitude decreasing sequence, are grouped in 10 latitude ranges. The numbers of coincident data pairs used to calculate the difference

profiles are indicated in the plots on a 3-km vertical interval. The maximum numbers of coincident data pairs used across the stations are listed in Table 1, as well. At both northern and southern higher latitudes, there are larger numbers of coincident pairs from about 20 up to > 100 compared to those at low latitudes, resulting from the ACE sampling pattern. Eureka is the station with the most coincident data points, 554 at ~15 km (resulting from an annual springtime validation campaign for ACE taking place at this location). In the tropics (Figs. 4f – 4h), the numbers of coincident

pairs are from a few to ~15. Despite the different numbers of coincident data pairs, the mean difference profiles show some consistent features across the stations. The upper altitude parts (>20 km) exhibit generally positive biases, increasing gradually with altitude from ~0 ppmv at around 20 km to about ~0.5 – 1.5 ppmv at about 30-32 km. These positive biases are mostly significant, as shown by the same sign of the values at the low and high ends of the standard error bars. For those ozonesonde stations reporting low ozone due to drop-offs, the effect would be high-biased

differences as shown in tropical stations, Hanoi, Costa Rica, and Ascension, while at Scoresbysund and Yarmouth, the differences are not distinct from those in other stations at the similar latitudes, and at Churchill, the differences are remarkably lower than those in other stations with unknown reason. There are also variabilities among the stations in the same latitude bands on the order of ~0.2 ppmv. Apart from those low-biased stations, these positive biases in the lower to middle stratosphere are consistent with the findings of Sheese et al. (2022) that earlier ACE v4.1/4.2 ozone

data are biased high versus other satellite datasets. In the troposphere (from ~5 km up to the tropopause), in the extra-tropics the mean differences exhibit high variability around ~0.04 ppmv across the stations (Figs. 4a - 4e, 4i, 4j), while in the tropics the biases are insignificant with variations around ~0.02 ppmv (Figs. 4f - 4h).

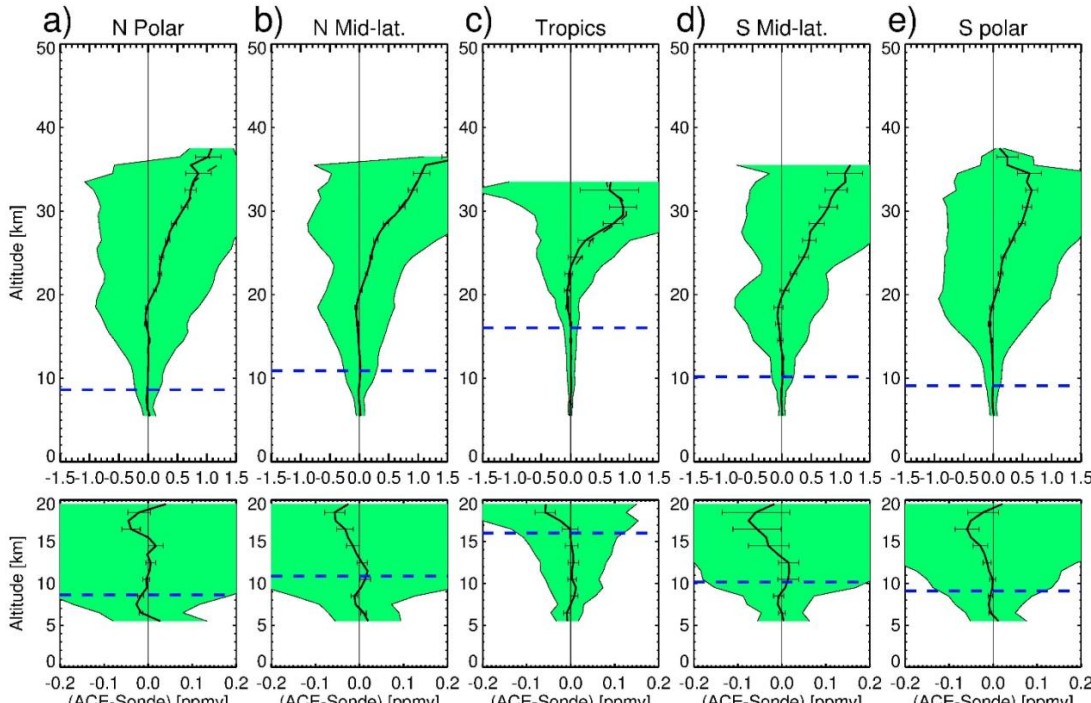

**Figure 5: The aggregated mean ozone profile differences in ppmv between coincident ACE-FTS and ozonesonde**
**measurements averaged over the stations for the latitude bands: the northern polar region (a, stations in Fig. 4a – 4b),**
**northern mid-latitudes (b, stations in Fig. 4c – 4e), the tropics (c, stations in Fig. 4f – 4h), the southern mid-latitudes (d,**
**stations in Fig. 4i) and the southern polar region (e, stations in Fig. 4j). The shaded areas represent $\pm 2\tilde{\sigma}$ ranges, $\tilde{\sigma}$ the**
**standard deviations over the aggregated data points, and the horizontal bars, $\pm 2\tilde{s}_{\tilde{\delta}}$, show the standard errors of the**
**aggregated means. The dashed lines in (a), (b) and (c) are the mean differences by including the drop-off stations (shown**
**only for reference purpose). The horizontal dashed blue lines are the average tropopause heights in the latitude bands**
**estimated from the temperature profiles in the ozonesonde data.**

Figures 5a – 5e are respectively the aggregated mean biases averaged over the stations in Figs. 4a - 4b for
the northern polar region, over the stations in Figs. 4c – 4e for the northern mid-latitudes, over the stations in Figs. 4f
– 4h for the tropics, over the stations in Fig. 4i for the southern mid-latitudes, and over the stations in Fig. 4j for the
southern polar region. In the averaging the six low-biased ozonesonde stations as mentioned above are excluded for
the final result (solid lines). For reference, companion calculations by including those six stations are shown with
dashed lines (Figs. 5a, 5b, 5c) and the overall impact of the low-biased stations is small. The latitude band averaged
biases show two distinct regions that differ by altitude. In the stratosphere, where there are mostly consistent positive
biases for ACE-FTS versus the ozonesondes, these biases start from negative values in the extra-tropics at about 17
km or near-zero in the tropics at about 22 km and then increase with altitude. At altitudes higher than ~32 km, as no



altitude cut-off was applied to the ozonesonde data, the bias assessment may be unreliable due to increasing ozonesonde measurement errors. In the UTLS regions the differences are at insignificant small amount of around ±0.02 ppmv.

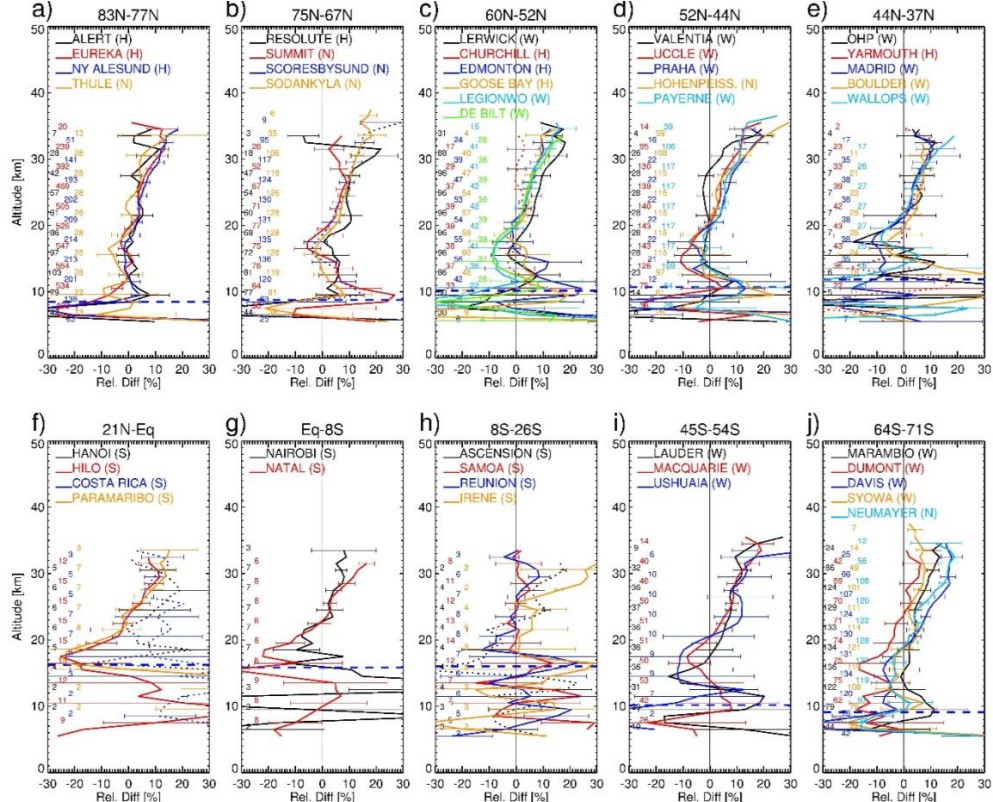

**Figure 6: Similar to Figure 4, but for the mean relative differences (%) between the ACE-FTS and ozonesonde profiles calculated from the coincident data pairs (ACE-sonde)/[(ACE+sonde)/2]. The horizontal bars are $\pm 2\, s_{\bar{\delta}}^{rel}$ , $s_{\bar{\delta}}^{rel}$ the standard errors of the estimated means.**

Figures 6a – 6j show the relative mean differences in percent (%) together with the standard error bars as

companion plots to Figs. 4a – 4j. As before, the six stations mentioned above are excluded from the discussion here. Because the relative differences are derived by dividing the absolute differences as shown in Figs. 4a – 4j by the mean ozone profiles, the upper parts show similar structures as in Figs. 4a – 4j, i.e., "tilted" differences across the stations, and increasing with altitude starting from ~20 km and reaching a maximum of 10% at ~30 km. In the lower stratosphere at altitudes around 20 km and a few kilometers below, particularly for the mid- and high latitudes (Figs.

6a – 6e, 6i - 6j), there are often negative differences seen for these ozonesonde stations. The large relative difference values in the troposphere (Figs. 6a – 6j) are attributed to the small tropospheric ozone concentrations. Figures 7a – 7e




present the ACE-FTS versus ozonesonde relative differences averaged over the five latitude bands. In the stratosphere, the bias structures show similar features to those in Figs. 5a – 5e at altitudes higher than 20 km. Below 20 km the absolute differences shown in Figs. 5a – 5e are amplified in the relative differences in Figs. 7a – 7e, specifically in the UTLS region where high variabilities are seen but no distinctive structures are seen in contrary to the plots derived from the monthly mean time series in Sect. 4.2.1.

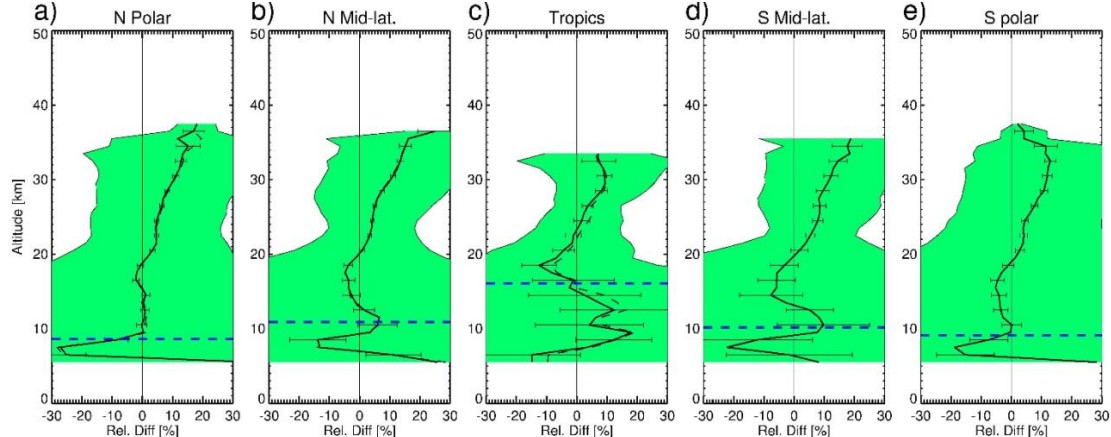

**Figure 7: Similar to Figure 5, but for the aggregated mean relative differences (%) between the ACE-FTS and ozonesonde profiles calculated from the coincident data pairs (ACE-sonde)/[(ACE+sonde)/2] in the five latitude bands. The shaded areas represent $\pm 2\tilde{\sigma}^{rel}$ ranges, $\tilde{\sigma}^{rel}$ the standard deviations over the aggregated data points, the horizontal bars, $\pm 2\tilde{s}_{\tilde{\delta}}^{rel}$, $\tilde{s}_{\tilde{\delta}}^{rel}$ the standard errors of the aggregated means.**

## 4.2 ACE-FTS versus ozonesonde monthly mean time series
### 4.2.1 Biases between ACE-FTS and ozonesonde monthly mean time series

This section reports the results of determining ACE-FTS ozone profile biases against ozonesonde data using the monthly mean comparison method. By choosing data within the latitude/longitude ranges of ±5°/±30° surrounding the ozonesonde stations, the ACE-FTS monthly mean time series were generated for all the ozonesonde stations. The absolute and relative differences between the ACE-FTS and ozonesonde monthly mean time series are calculated using the method in Sect. 3.1 and 3.2, where the data pairs used are the monthly mean time series for the common months. Here only the relative differences are shown in Figs. 8a – 8j, where the numbers of monthly mean points used for difference calculations are indicated and the maximum numbers in the vertical profiles are also listed in Table 1. These are generally larger than the numbers of coincident data pairs used (as shown in Figs. 6a – 6j), especially in the tropical regions. Relative to the coincident data analysis, the bias determination using monthly mean time series employs more relaxed coincidence criteria with a possible outcome that averaging over more samples to produce the monthly means likely outweighs larger geophysical variability. The benefit can be seen below.

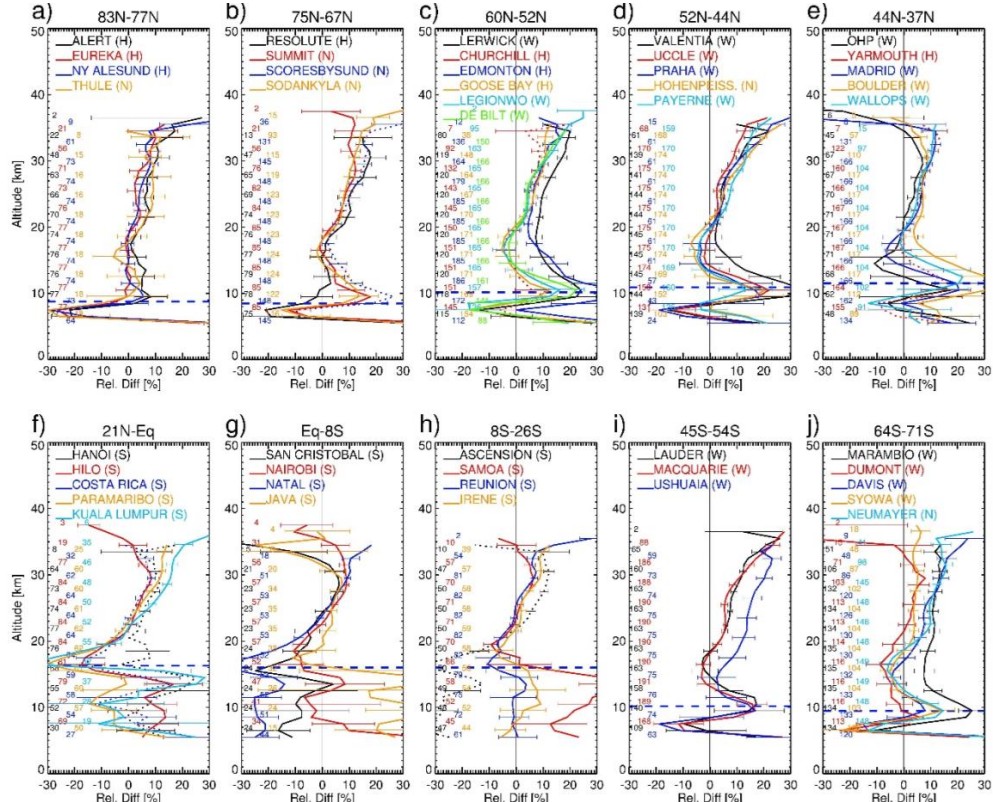


**Figure 8: Similar to Figure 6 but for the mean ACE-FTS and ozonesonde relative difference profiles in percentage (%) estimated from their respective monthly mean time series at all the stations.**

The difference profiles shown in Figs. 8a – 8j are generally consistent with those in Figs. 6a – 6j in terms of the altitude-dependent positive biases seen in the upper altitude parts of the profiles. In particular, the time series difference profiles at mid-latitudes in Europe (Figs. 8c – 8e) and in tropics (Figs. 8f – 8h) exhibit more smooth and consistent features than those in Figs. 6c – 6e and in Figs. 6f – 6h, respectively. In the troposphere, the time series difference profiles (e.g., Figs. 8c - 8e) are also more consistent than those for coincident pairs in Figs. 6c - 6e. This is attributed to more points utilized in the monthly mean time series for the comparison. For three sites in the tropics,

namely Kuala Lumpur, San Cristobal, and Java, coincident pair comparisons were not obtained. However, monthly mean time series comparisons are possible and appear in Figs. 8f and 8h. At Churchill the difference profile in Fig. 6c, calculated from 39 coincident pairs, is distinct from other profiles, while in Fig. 8c, with 150 months used, the Churchill profile is similar to the other profiles at comparable latitudes. The latitude band averaged mean difference profiles (Figs. 9a – 9e) are also calculated from the ACE-FTS and ozonesonde monthly mean time series. The positive

biases seen in the stratosphere are very similar to those seen from the coincident pair comparisons in Figs. 7a – 7e, while in the UTLS region the averaged differences show more consistent and structured features. One remarkable feature is that one of the profile "turning points" is coincident with the tropopause heights at all the latitude bands.





The differences shown in Figs. 9a – 9e at the tropopause altitudes are negative by about -15% in the tropics, and positive by 7 – 20% in the extra-tropical regions. This is an interesting feature showing the differences appear to depend on the atmospheric temperature profiles. The method of using monthly mean time series for the bias estimation as presented in this section demonstrates it is an alternative way to the conventional coincident data analysis and is particularly useful when strict coincidence criteria do not yield sufficient data points for comparisons.

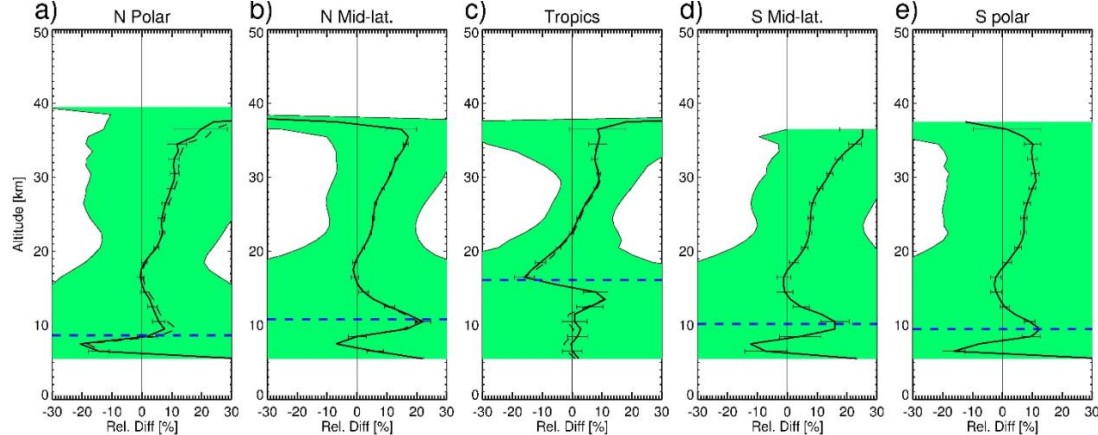

**Figure 9: Similar to Figure 7, but for the aggregated mean relative ozone profile differences in percentage (%) between the ACE-FTS and ozonesonde monthly mean time series.**

### 4.2.2 Variation patterns and mean states in the ACE-FTS and ozonesonde monthly mean time series

Figure 10 displays the ACE-FTS and ozonesonde monthly mean ozone VMR time series for their common months at selected altitudes in the lower stratosphere from 2004 to 2023. These are shown for 16 representative ozonesonde stations in each 10°-wide latitude band. Note, there is no station between 30°S - 40°S. The altitudes are selected such that temporal variations are predominant with the distinct periods attributed to atmospheric dynamics. The monthly samples as shown in Fig. 10 are not evenly distributed across the latitudes. In the Arctic region (e.g., Eureka and Resolute), the common ACE-FTS and ozonesonde monthly samples appear only in two seasons, northern spring (February – March) and northern fall (August – October). At mid-latitudes in the northern hemisphere (NH), e.g., Lerwick, and Goose Bay and in the southern hemisphere (SH), e.g., Lauder and Macquarie, the monthly samples appear to be dense, as is also seen in the Antarctic region (e.g., Davis and Neumayer). In the tropics, the monthly samples are much less dense than those at higher latitudes, but there are samples available in every season.





Figure 10: The ACE-FTS (black) vs. ozonesonde (red) monthly mean time series at selected altitudes for the stations from north to south (from top to bottom panels). The vertical bars represent $\pm 2\sigma$, $\sigma$ the standard deviations from those values which generated monthly means.



The ACE-FTS and ozonesonde time series in Fig. 10 appear to have very good agreement at mid- and high-
latitudes in both the NH (e.g., Eureka, Resolute, Lerwick, Goose Bay, Payern, Wallops) and SH (e.g., Lauder,
Macquarie, Davis, and Neumayer) with the annual cycle as the dominant variation pattern.  The variations in the NH
and SH show the expected six-month phase difference. In the tropics, the ACE-FTS and ozonesonde time series agree
well except at Hanoi.  In the equatorial region, it is not the annual cycle but other temporal patterns such as the QBO
(e.g., Nairobi (1.3°S, 36.8°E), Paramaribo (5.8°N, 55.2°W)) that are the apparent dominant patterns. In general, the
ACE-FTS time series at these stations and altitudes agree very well with the ozonesonde monthly mean time series.
Figure 11 is the altitude/station section of the correlation coefficients between the ACE-FTS and ozonesonde monthly
mean time series (including those six drop-off stations) with the horizontal axis in the order of the station latitudes.
High correlations (≥0.75) are present at the altitudes where the time series are characterized by prominent temporal
variation patterns. These are located above the tropopause and extend to higher altitudes up to ~30 km and even 35 km
in the polar regions. There are some areas where the correlations appear to be poorer, with values of ~0.5 such as in
the tropical stratosphere (27 – 32 km), and northern mid-latitude stratosphere (25 – 28 km) as well as at some stations,
e.g., Ushuaria, Samoa, Asension, Java, Hanoi, and Prague. Those variations with good agreement are attributed to the
dynamic forcing on the ozone field, as ozone is a tracer gas in the lower stratosphere where ozone changes are
primarily resulting from transport rather than photochemistry-induced ozone creation or destruction (Toihir et al.,
2018). Several researchers have used multivariable linear regression methods to extract typical atmospheric modes
from ozone measurements. The QBO, annual oscillation, and semi-annual oscillation modes, etc. have been derived
from MIPAS global ozone data up to 50 km (Eckert et al., 2014); from SHADOZ ozonesonde data for the tropics
(Toihir et al., 2018); and from ozonesonde data at Lauder for representing the mid-latitude southern hemisphere (Zeng
et al., 2017). The amplitudes and significance of these modes are a function of altitude and latitude. With the general
good agreement and high correlations between ACE-FTS and ozonesonde data, similar analyses can be done using
the ACE-FTS monthly mean time series data.

Figure 12 shows the averaged values and standard deviations calculated from the ACE-FTS and ozonesonde
monthly mean time series (including the six drop-off stations) displayed on an altitude grid with an interval of 3 km.
This plot was made in a similar way to the comparison study between Aura-MLS and ozonesonde data (Jiang et al.,
2007). The averaged values represent the mean state of ozone, which is a function of altitude and latitude, and the
standard deviations give the variability of the ozone changes such as those presented by the time series shown in Fig.
10. Both the ACE-FTS and ozonesondes show similar mean ozone patterns characterized by high values in the tropical
mid-stratosphere (~30 km) where ozone is generated, and the latitude distribution driven by transport via the BDC,
which is poleward and directed downwards in the extra-tropical regions. The tropical area in the UTLS, between
9.5 km and 21.5 km, exhibits low values of ozone, as this is where ozone is least accumulated. There are clearly
discrepancies in that the ACE-FTS mean values are larger than the ozonesonde values across the stations. This occurs
at higher altitudes, such as above 21.5 km, and is consistent with the features revealed in Figs. 3, and 4, for example.
Regarding ozone variability together with the measurement uncertainty, the ACE-FTS variabilities in the troposphere
(i.e., at 6.5 km and 9.5 km in Fig. 11) exhibit larger values than those of the ozonesondes. In the lower stratosphere,



both the ACE-FTS and ozonesondes exhibit variabilities of similar magnitudes. In regions of low ozone concentration, such as the tropical troposphere, the variabilities from both ACE-FTS and ozonesondes are lower than those in the other regions.

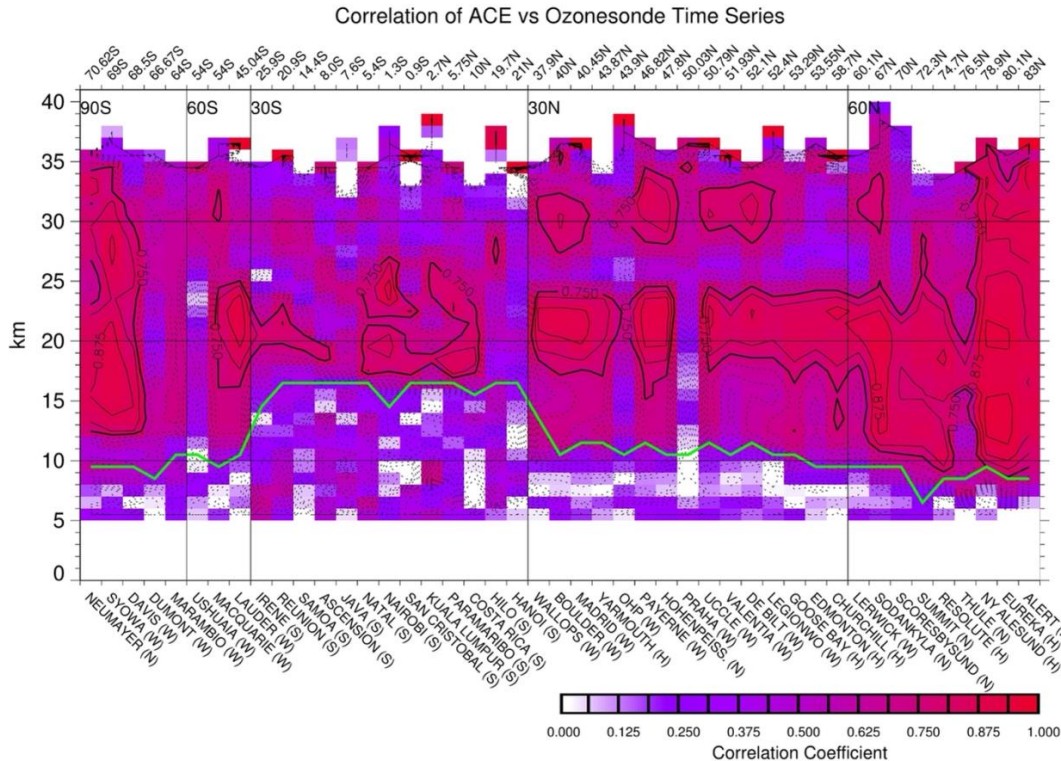


**Figure 11: The correlation coefficient profiles between the ACE-FTS and ozonesonde monthly average time series for all the stations. The horizontal axis denotes the ozonesonde station latitudes as shown on the top of the panel with the station names shown at the bottom. The contour isolines have an interval of 0.05 and the solid (rather than dotted) isolines represent the high correlation coefficients ≥ 0.75. The green solid line separating the troposphere and stratosphere shows**

**the average tropopause height line across the stations.**



**Figure 12: The averaged values and $\pm 2\sigma$ vertical bars, $\sigma$ the standard deviations calculated from the ACE-FTS (black "x") and ozonesonde (red "x") monthly mean time series displayed on 3 km altitude grid. The horizontal axis denotes the station latitudes (given at the top) with the station names at the bottom.**




### 4.3 Instrument drifts relative to the ozonesonde measurements

The ACE-FTS instrument drift is estimated as the linear trend of the differences between the ACE-FTS and
ozonesonde de-seasonalized time series and was calculated at all ozonesonde stations (including the six drop-off
stations) listed in Table 1. Although ozone trend estimation is not the focus of this study, it is helpful to keep in mind
that the underlying ozone concentrations in the lower stratosphere (< 35 km) during the study period (2004 – 2022)
are generally small with insignificant trends after 2000 (e.g., Steinbrecht et al., 2017, Tarasick et al., 2016,
SPARC/IO3C/GAW, 2019). The ACE-FTS ozone measurement drifts in the v4.1/4.2 data have been assessed by
comparing with other satellite data at 15 - 40 km for the period 2004-2021 (Sheese et al., 2022). This study serves to
extend this validation work to lower altitudes covering between 5 - 33 km.

Figures 13a – 13j show the instrument drifts in ppmv dec$^{-1}$ at all the ozonesonde stations over the period 2004
- 2023, with dotted lines for the drop-off stations (flag 1 in Table 1) and solid lines for non-drop-off stations (flags 2,
3 in Table 1). The plots at low altitudes (< 20 km) with enlarged graphs are also displayed. Figures 14a – 14j are the
corresponding plots for the relative instrument drifts in % dec$^{-1}$. To consider the impact of the polar vortex as
mentioned in Sect. 3.7, additional analyses were carried out by reducing the ACE-FTS data sampling areas by setting
the longitudinal range to ±10° and removing data in northern polar springtime (Jan. – Mar.) north of 65°N and southern
polar springtime (Sept. – Nov.) south of 65°S. The results obtained by removing possible polar vortex data did not
change the general features in the drift and uncertainty plots. At some stations such as Eureka, restricting data to those
outside of the polar vortex resulted in no ACE-FTS samples at all.



**Figure 13: The instrument drifts in ppmv dec$^{-1}$ derived from the linear trends of the differences between the ACE-FTS and ozonesonde de-seasonalized monthly mean ozone profile time series at the ozonesonde stations. As before, the dotted profiles are those from drop-off stations. The $\pm 2\,s_{\hat{b}}$ standard error bars are given at the 3 km vertical interval. The horizontal dashed blue lines are the average tropopause heights in the latitude bands estimated from the temperature profiles in the ozonesonde data.**




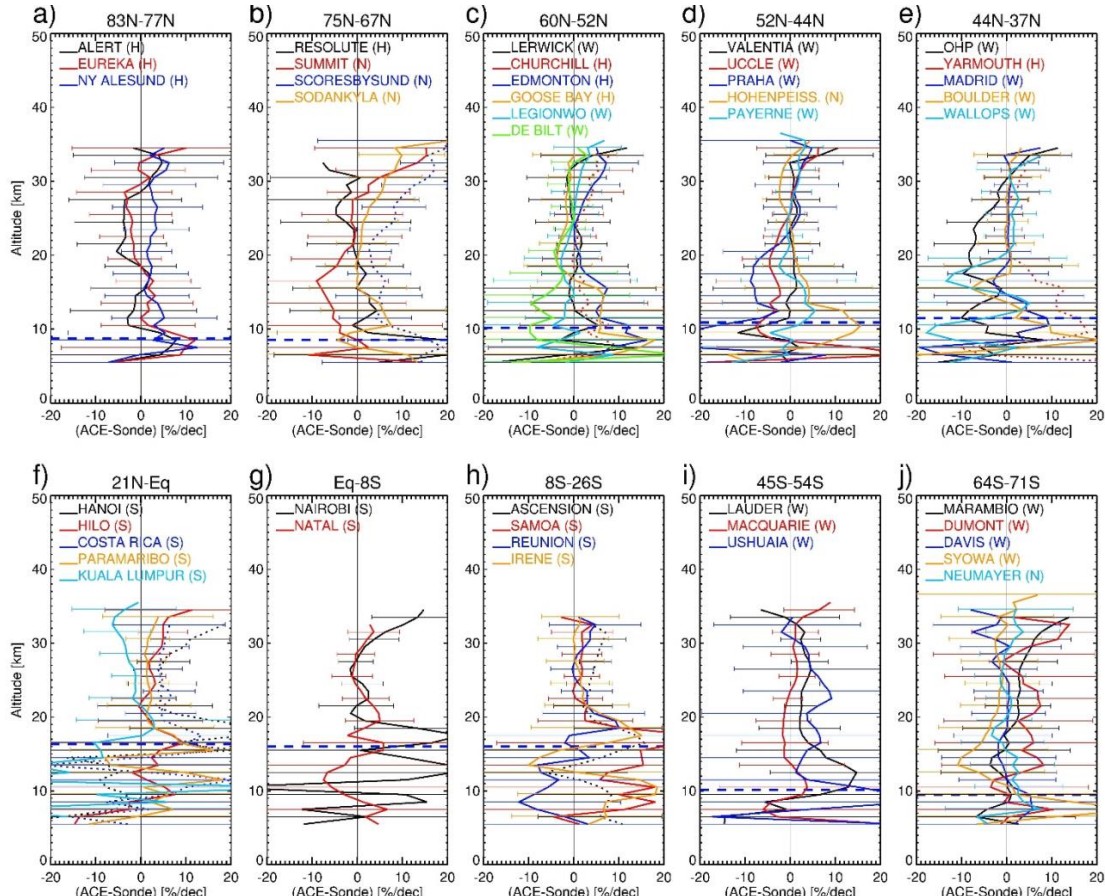

**Figure 14: Similar to Figure 13, but for the instrument drift calculated as relative differences in % dec⁻¹.**

The plots for the drop-off stations as shown in Figs. 13 and 14 (dotted profiles) are included for reference and evaluation purposes. At Churchill and Yarmouth, the drifts are not different from the other drift profiles at the similar latitudes. At Scoresbysund, Costa Rica and Ascension, the positively biased drifts may be caused by the drop-offs.

        At the northern high latitudes (> 65°N) (Figs. 13a, 13b and Figs. 14a, 14b), the instrument drifts at individual

stations are around -5 – 2% dec⁻¹ with changing signs at different stations and with large uncertainties of the order of ±10% dec⁻¹, and at Summit the drifts can be as large as -10% below 20 km with the uncertainties of ±20% dec⁻¹. The high variability in the polar region is consistent with the study of Tarasick et al. (2016) that showed the linear trends derived from the Canadian ozonesonde data can change signs between positive and negative. This is especially the case when the underlying long-term ozone trends are small.

Figures 13c - 13e and Figs. 14c – 14e present the instrument drifts at northern mid-latitude ozonesonde stations (37°N – 60°N). Most of them are in Europe and show generally consistent features with drifts within ±2%





dec$^{-1}$ and uncertainties of ±5% dec$^{-1}$ for altitude ranges around 20 – 33 km except at OHP which has seemly unrealistic negative drifts of -10% at about 18 – 25 km.

        In the tropics (Figs. 13f - 13h and Figs. 14f - 14h), the instrument drifts in general exhibit positive drifts at
0 - 5% dec$^{-1}$ above 20 km with magnitudes that vary with altitude except at Kuala Lumpur where the negative drifts are seen with values between -2 and -8 % dec$^{-1}$ above 22 km. Below this altitude, the relative drifts are characterized by larger values at ±20% dec$^{-1}$, but are insignificant due to large uncertainties. As analyzed by Thompson et al. (2021) using SHADOZ ozonesonde data in the tropical troposphere, ozone trends are characterized by high regional, seasonal and altitude variability attributed to dynamics. This feature together with the low ozone amounts in the tropical
troposphere result in high relative drifts and variabilities in % dec$^{-1}$ and small absolute drifts in ppmv dec$^{-1}$.

        In the southern hemisphere there are no stations between 30°S and 40°S and in the mid-latitudes (Fig. 13i and Fig. 14i), there are three stations, Lauder, Macquarie, and Ushuaria. The instrument drifts at Lauder and Ushuaria show insignificant positive drifts of about 2 – 8% dec$^{-1}$ above the tropopause, while at Macquarie the instrument drifts are around ±1% dec$^{-1}$.

In the Antarctic region the instrument drifts vary among stations (Fig. 13j and Fig. 13j), just as those in the Arctic region. This high variability is attributed to the strong seasonality of Antarctic ozone and the interannual variation of this seasonality as found by Sepúlveda et al. (2021) in the difference time series between the OMPS-LP ozone profiles and the Antarctic ozonesonde data.

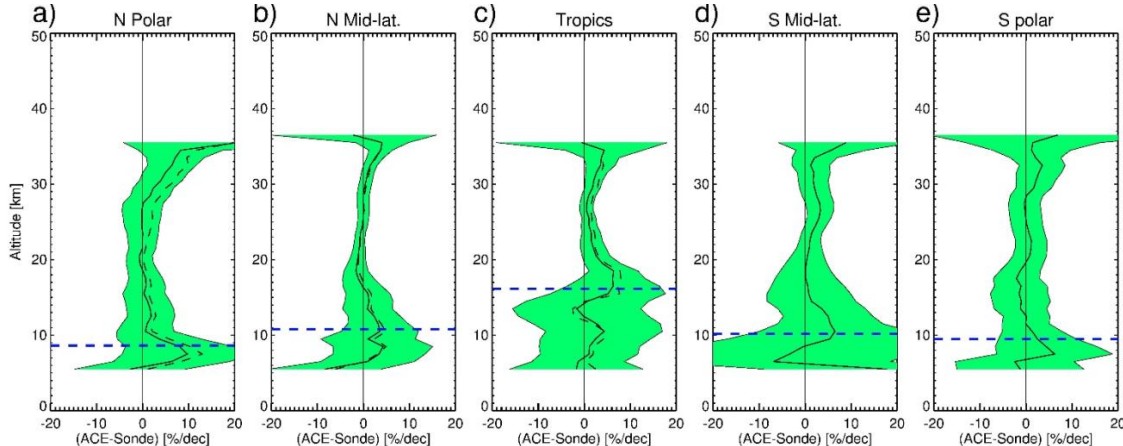


**Figure 15: The aggregated mean instrument drifts in % dec$^{-1}$ averaged over the ozonesonde stations excluding those with drop-offs (solid lines) at five latitude bands for the northern polar region (a), for northern mid-latitudes (b), for the tropics (c), for southern mid-latitudes (f), and for the southern polar region (e). The dashed lines are the mean instrument drifts from the entire set of stations including those drop-off stations (a, b, c). The stations aggregated for each region are the**

**same as those indicated in Fig. 4. The shaded areas represent ±2 $\widetilde{S}_{\tilde{b}}^{\,rel}$ , the standard errors of the aggregated mean drifts.**

**The horizontal dashed blue lines are the average tropopause heights in the latitude bands estimated from the temperature profiles in the ozonesonde data.**





Figures 15a – 15e present the aggregated mean ACE-FTS instrument drifts averaged over the five latitude bands by excluding those drop-off stations (solid lines). For reference, the aggregated mean drifts obtained by including all ozonesonde stations are shown as well (dashed lines in Figs. 15a, 15b and 15c) – small changes at 1 % dec$^{-1}$ compared with the solid lines. In the northern polar region and mid-latitudes (Figs. 15a, 15b), the aggregated mean instrument drifts show small insignificant drifts within ±1% dec$^{-1}$. In the tropics (Fig. 15c) and the southern mid-latitudes (Fig. 15d), positive drifts at 0 - 3% dec$^{-1}$ varying with height are seen at most altitudes above the tropopause. In the southern polar region, the averaged ACE-FTS instrument drifts exhibit small insignificant drifts at ±1% dec$^{-1}$. The small values of the aggregated drifts are attributed to averaging over several ozonesonde stations available in the latitude bands, although the individual station drifts feature high variabilities.

## 5 Conclusions

This study validated ACE-FTS v5.2 ozone data using ozonesonde data from 45 stations from four global networks, NDACC, WOUDC, SHADOZ, and HEGIFTOM. As part of this study, the ozonesonde network data were examined for drop-off features using ACE-FTS data and they were found to be consistent with the results obtained by Stauffer et al. (2022). These six affected sonde stations were not included in further analysis. The biases between the ACE-FTS and ozonesonde profiles were examined by analyzing coincident data pairs. In general, the biases are small at lower altitudes and increase with altitude up to 10% at the higher altitudes (just below the balloon burst altitudes). This is consistent with the findings that the earlier ACE-FTS v4.1/4.2 ozone data are larger than other satellite instruments in the mid-stratosphere (Sheese et al., 2022). This bias structure is consistent across all the ozonesonde station latitudes.

This study also presents the result of using the monthly mean comparison method. The biases between the ACE-FTS and ozonesonde monthly mean time series exhibit similar bias patterns to those derived from the coincident data analysis. Moreover, these monthly mean time series comparisons display smoother curves and achieve more consistency among stations at similar latitudes than the coincident pair analysis, and in the UTLS region show structured features correlated with the tropopause altitudes. The ACE-FTS time series can also capture distinct temporal variation patterns such as annual variability and QBO at certain latitudes and altitudes, display the mean state of ozone, and are highly correlated with the ozonesonde time series at certain altitudes. All these results support the feasibility of the method presented here for generating and validating ACE monthly mean ozone time series.

The ACE-FTS instrument drifts are assessed at all individual stations for 2004 – 2023 and the drifts determined in the same latitude band are compared individually. These show variability among the stations in the same latitude band, except at northern mid-latitudes (a cluster of European sites) where individual drifts are mostly consistent. Averaging the individual drifts for each latitude band results in small insignificant drifts for stratospheric ozone within ±1% dec$^{-1}$ in the polar regions (Arctic and Antarctic), and < 1% dec$^{-1}$ at northern mid-latitudes, positive drifts of 0 - 3% dec$^{-1}$ in the tropics and at the southern mid-latitudes. In the troposphere, the ACE-FTS instrument drifts, aggregated by latitude band, vary with altitude and are generally within ±10% dec$^{-1}$ with uncertainties of 10 % dec$^{-1}$.





## Appendix A    Examination of "drop-off" features with ACE-FTS data

The drop-off analyses using ACE-FTS data were performed only for the stations operated with EnSci ECC ozonesondes (described as Z models in the metadata), as drop-offs occurred only in this type of ozonesondes with specific production lots (serial numbers > 25000) and correlated with the associated changes in the characteristics of the mechanical pumps (Stauffer et al., 2022; Nakano and Morofuji, 2023). For another type of ECC ozonesondes manufactured by Science Pump Corporation (SPC) (described by 6A model in the metadata) no drop-offs are found (Stauffer et al., 2022). The ozonesonde data used here are re-processed recently through the homogenization procedure and have achieved substantial improvements in drop-offs for most of affected stations (Stauffer, et al., 2022). Using ACE-FTS data to check the drop-offs is to provide another independent examination from the perspective of employing a solar occultation satellite instrument which has much less sampling density than those used in Stauffer et al. (2020, 2022) for identification and quantification purpose, e.g., Aura-MLS, Aura-OMI (Ozone Monitoring Instrument) and OMPS-LP.

To investigate the drop-off features, the differences between the ozonesonde and ACE-FTS de-seasonalized monthly mean time series are examined. The differences largely remove their respective annual cycles and other common temporal components, e.g., QBO, ENSO, leaving the long-term drifts and residuals. Dividing the differences by the corresponding ozonesonde monthly mean values, the relative differences are calculated. For better visualization of the difference profile time sequence (see Fig. A1 and figures in Supplement), the monthly mean time series in absolute value (ppmv) were further re-binned into seasonally averaged time series. The 3-month re-binning is based on the consideration that ACE-FTS monthly mean time series often have missing data in certain months. Re-binning results in new time series with reduced samples but having more even distribution of the data in time for easy display.

To quantify the drop-off, the partial column ozone (PCO) change between the potential drop-off period and the prior period since 2004 (non-drop-off period) is estimated. The drop-off periods are specified based on the serial numbers of EnSci sondes (S/N > 25000), which are usually provided in the ozonesonde metadata. To use PCO instead of TCO is based on the consideration that at lower altitudes the ACE-FTS ozone data are either not available or exhibit high variability, while drop-offs usually occur at above 50 mb (~20 km) (Thompson et al., 2021; Stauffer et al., 2022). From their respective data of ozone VMR, atmospheric pressure and temperature the ozonesonde and ACE-FTS difference profiles for the de-seasonalized number density time series are first calculated. The mean number density difference profiles, averaged over the potential drop-off period and the prior non-drop-off period are calculated, respectively. Integrations of the two profiles from 15 to 25 km give two PCO differences. The difference between the two PCO differences, divided by the ozonesonde POC averaged over the entire period, gives the drop-off estimate.

For drop-off analyses 24 stations operated with the EnSci sondes were selected (Table A1). The estimated drop-off along with the number of months and the period of S/N > 25000 are given for each station in Table A1. The ozone profile difference time series at these stations are shown in the Supplement. As a representative example, Fig. A1 shows the relative difference profile time series (in percent) between ozonesondes and ACE-FTS at Yarmouth. The drop-off features are visible as persistent negative values as step changes in the lower stratosphere around 2016 - 2020, approximately consistent with the period of 2015/4 – 2019/10 when S/N > 25000. After 2019/10 drop-off is no longer considered as the sonde model switches from EnSci to SPC. To identify a drop-off station a step change in



TCO larger than 3-4% was used in Stauffer et al. (2020, 2022). Here, if a PCO drop is larger than 3 – 4%, the station is regarded as a drop-off station. In Table A1, the drop-off stations are labelled with flag 1. They are primarily at Canadian sites, Churchill (-4.3%), and Yarmouth (-4.4%), at Greenland's Scoresbysund (-5.6%), and at the tropical sites, Ascension (-4.4%), Costa Rica (-3.4%) and Hanoi (-4.4%). Stations with flag 2 are those having drop-offs less than 3% and are deemed not affected by drop-offs.

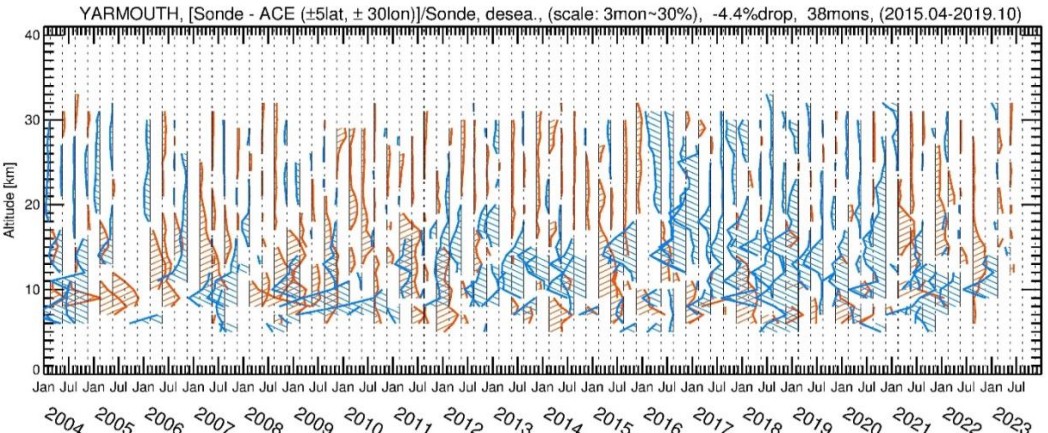

**Figure A1: Seasonally averaged relative differences (%) between the ozonesonde and ACE-FTS de-seasonalized monthly mean time series at Yarmouth, Canada. The red/blue colors indicate the positive/negative differences, respectively. A scale of 30% relative difference is represented by the distance between the two adjacent dashed lines (3 months). The drop-off amount, the period and the months of operating EnSci sondes with S/N > 25000 are indicated at the top.**

| Station | Drop-off in percent, number of months and period of EnSci sonde S/N > 25000 | Drop-off Flag |
|---|---|---|
| Alert | -0.3%, 36 months, (2014/8 – 2023/7) | 2 |
| Eureka | -1.9%, 46 months, (2016/1 – 2021/3) | 2 |
| Resolute | 0.4%, 54 months, (2013/4 – 2021/1) | 2 |
| Summit | -2.3%, 43 months, (2013/11 – 2017/7) | 2 |
| Scoresbysund | -5.6%, 101 months, (2013/12 – 2023/2) | 1 |
| Sodankylä | -1.9%, 21 months, (2016/1 – 2019/10) | 2 |
| Churchill | -4.3%, 24 months, (2016/6 – 2021/1) | 1 |
| Edmonton | -1.2%, 63 months, (2015/7 – 2023/5) | 2 |
| UCCLE | 1.9, 83 months, (2015/9 – 2022/7) | 2 |
| OHP | 3.9%, 21 months, (2020/1 – 2021/12) | 2 |
| Yarmouth | -4.4%, 38 months, (2015/4 – 2019/10) | 1 |
| Boulder | -1.7%, 33 months, (2019/5 – 2022/6) | 2 |
| Hanoi | -4.4%, 60 months, (2016/3 – 2022/9) | 1 |





| Hilo | -1.2%, 96 months, (2013/10 - 2022/9) | 2 |
|---|---|---|
| Costa Rica | -3.4%, 73 months, (2014/7 – 2021/12) | 1 |
| Kuala Lumpur | 1.3%, 74 months, (2014/12 – 2021/12) | 2 |
| Nairobi | 0.2%, 69 months, (2014/4 – 2020/3) | 2 |
| Ascension | -4.4%, 67 months, (2016/3 – 2022/9) | 1 |
| Samoa | -2.1%, 90 months, (2013/9 – 2022/8) | 2 |
| Reunion | -1.4%, 41 months, (2015/6 – 2018/12) | 2 |
| Lauder | -1.7%, 33 months, (2019.5 – 2022/6) | 2 |
| Marambio | -2.0%, 60 months, (2014/12 – 2019/11) | 2 |
| Syowa | -0.6%, 93 months, (2015.2 – 2022.12) | 2 |

**Table A1: The drop-off analysis result for those stations operated with EnSci ozonesondes. Listed are the drop-off in percent, number of months and the period of the EnSci sondes with S/N larger than 25000. Flag 1 is indicated for drop-off larger than 3%, and flag 2 for less than 3% or even positive.**



*Data availability.* The data used in this study are all available publicly.

The ACE-FTS v5.2 ozone data were downloaded from the ACE data archive at the University of Waterloo: https://databace.scisat.ca/level2/ace_v5.2 (registration required).

The ACE-FTS data quality flags for ACE-FTS Level 2 version 5.2 data set are available from: https://doi.org/10.5683/SP3/NAYNFE.

The ozonesonde data were downloaded from the following sources:

- Network for the Detection of Atmospheric Composition Change (NDACC): https://www-air.larc.nasa.gov/missions/ndacc/data.html;
- World Ozone and Ultraviolet Data Centre (WOUDC): https://woudc.org/;
- Southern Hemisphere Additional OZonesondes (SHADOZ): http://tropo.gsfc.nasa.gov/shadoz/index.html;

- Harmonization and Evaluation of Ground-Based Instruments for Free Tropospheric Ozone Measurements (HEGIFTOM): https://hegiftom.meteo.be/.

*Contribution of the authors.* JZ performed and coded the analyses and wrote the manuscript. KW initiated the project, coordinated with the co-authors, and helped to edit and finalize the manuscript. PS produced the ACE-FTS data quality

flag files. CB produced the ACE-FTS ozone data. RS shared his expertise to ensure the study in line with the progress in the field. AMT provided the insightful suggestions for improving the manuscript. DT provided the expertise to analyse the ECC data with the Canadian ozonesonde data.

*Competing interest.* The authors declare that there are no conflicts of interest involved.


### Acknowledgement

This study was funded by a contract from the Canadian Space Agency (CSA). The Atmospheric Chemistry Experiment (ACE) is a Canadian-led mission mainly supported by the CSA. We thank Peter Bernath for his leadership of the ACE mission. The authors thank the Principal Investigators, organizations/institutions that create and maintain

the data sets at NDACC, WOUDC, SHADOZ and HEGIFTOM.



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
