# Peer review of "Validation of ACE-FTS version 5.2 ozone data with ozonesonde measurements"

_EGUsphere, 2024_

## Author Response (AR1)

**Letter to the Editor – "Validation of ACE-FTS version 5.2 ozone data with ozonesonde measurements" by Jiansheng Zou et al.**

We thank the two anonymous reviewers for their comments. The reviewers' comments are given below with our responses provided in light blue indented text.

In addition, we have made a few changes to the paper based on new data provided by some of our colleagues and continued discussions among the co-authors.  These additional updates are as follows. Also, we fixed some typographical errors and made a few clarifications.

These include:

Instances of "insignificant" have been changed to "non-significant" throughout the paper.

One new station has been added in the southern mid-latitudes and two data sets have been updated:

- The Broadmeadows station in Australia now has additional data available so that it can meet the 10-year time series cut-off in our site selections.  This is the only station at 30S-40S which augments the analysis coverage.
- Reprocessed Scoresbysund data were recently made available from the HEGIFTOM project. The analysis with these new data shows that the low biases, existing in the comparisons with the previous data sets from NDACC, have been removed. Also, it is no longer identified as a drop-off station in our analyses.
- The source for the Wallops Island station data was switched from WOUDC to SHADOZ to obtain a longer time period for analysis.

Also, the co-authors have had further discussions on how to identify the drop-off stations in the tropics. The final decision is to remove both the drop-off stations identified in Stauffer et al. (2022) and those stations identified in this study. As ACE sampling is sparse in the tropics, we are being conservative to give more confidence to the final calculations for biases and drifts. The analysis of drop-off stations using ACE is now identified in the abstract and in a few places some clarifications have been made about these stations.

Because of these changes, all the plots in the paper have been updated.

**Reviewer Comment 1**: 'Thorough validation paper, maybe a bit detailed and lengthy', Anonymous Referee #1, 25 Jul 2024

The paper reports on the validation of ACE-FTS version 5.2 ozone profile data using ozonesondes. The underlying analysis is thorough and correct.

The main conclusions from the paper are that ACE-FTS ozone profile data agree with sonde data around 20 km altitude, are about 10% higher than sonde data at 30 km. At the extratropical tropopause, ACE-FTS ozone data tend to be 5 to 15% higher than the sondes. Temporal ozone variations over the last 20 years are seen very similar by ACE-FTS and by the sondes. There are no significant drifts between the sonde and ACE-FTS time series. The paper is generally well written, although I find it somewhat lengthy and detailed. Nevertheless, I think it is acceptable in its present form.

I have only two minor suggestions for changes:

In Figure 10, I am wondering why the same data gaps appear for the ozone sondes and ACE-FTS. This is most notable at Irene where the gap goes from 2008 to 2013. Are really the same months missing in both data sets? I think this is probabably a plotting error and should be fixed.

> We note that this is not a plotting error.  Figure 10 shows only the monthly mean data points for the months when both ACE-FTS and ozonesonde have valid data (e.g., "their common months" as described in the text). For Irene, there are no ozonesonde data available between 2008 and 2012 (see SHADOZ database:  https://tropo.gsfc.nasa.gov/shadoz/Irene.html), although ACE-FTS has valid data for this period.
>
> A sentence has been inserted in the figure caption for Fig. 10 to clarify this point. "The data points are at the common months when both ACE-FTS and ozonesondes have valid monthly means."

In Figure 15, I find the green area misleading. In all the previous plots, the green area denoted the standard deviation range of the individual station results, whereas error bars denoted the standard error of the mean. Different from that, the green region in Fig. 15 denotes the standard error of the mean drift. For consistency with the previous plots, I suggest to change that and use error bars, instead of the green area.

> We agree with this suggestion and have changed the plot as described.

**Reviewer Comment 2**: 'Comment on egusphere-2024-1916', Anonymous Referee #2, 02 Aug 2024

This paper addresses the quality of ACE-FTS v5.2 ozone, and instrument drifts by comparing with correlative ozonesondes between upper troposphere/lower stratosphere (UTLS) and mid-stratosphere (~30 km). The overall scientific approaches and technical methods are sound. The research results are also useful to the community. There are, however, some issues/questions that need to be addressed or clarified before publication.

The following are specific comments and questions for this paper

1.  Line 20. "The ACE-FTS ozone profiles exhibit in general high biases in the stratosphere, increasing with altitude up to ~10% at around 30 km, …". The ACE-FTS ozone profile does not show systematic high bias in the stratosphere. The positive biases are mainly for altitudes above ~20 km. For altitudes below 20 km and a few kilometers above tropopause, it tends to show negative biases up to ~5 -10% (Figure 7). To make it clear, I suggest modifying the original text to "The ACE-FTS ozone profiles exhibit in general high biases in the stratosphere for altitudes above 20 km, increasing with altitude up to ~10% at around 30 km, …", or other similar sentences.

Rewording has been done for this line.

Original sentence: "The ACE-FTS ozone profiles exhibit in general high biases in the stratosphere, increasing with altitude up to ~10% at around 30 km, and have local maximum differences with ozonesonde profiles at the tropopause heights"

Reworded sentence: "The ACE-FTS ozone profiles exhibit in general high biases in the stratosphere for altitudes above ~20 km, increasing with altitude up to ~10% at around 30 km. For altitudes between 20 km and the tropopause, the biases of up to ±10% are found, depending on altitude and latitude with the largest biases found in the tropics and southern mid-latitudes."

2.  Line 24, "…., and a small positive insignificant drift of 0 - 3 % dec$^{-1}$ in the tropics and southern mid-latitudes with overall uncertainties at 2 – 3 % dec$^{-1}$ (2σ level) in the low stratosphere". The above sentence needs to be clarified. In Figure 15, it shows positive drifts of ~5% dec$^{-1}$ in the tropic and southern mid-latitude lower stratosphere (i.e. ~16 -19 km in the tropics, ~10-14 km in southern mid-latitude), with uncertainties of ~5-10% dec$^{-1}$. The results in Figure 15 are different from the above texts in line 24.

This sentence has been reworded as follows.

Original sentence: "Averaging the individual station instrument drifts within several latitude bands results in small insignificant drifts of within ±1 % dec$^{-1}$ in the northern mid- to high latitudes, and the southern high latitudes, and a small positive insignificant drift of 0 - 3 % dec$^{-1}$ in the tropics and southern mid-latitudes with overall uncertainties at 2 - 3 % dec$^{-1}$ (2σ level) in the low stratosphere.

Reworded sentence: "Averaging the individual station instrument drifts within several latitude bands results in small insignificant drifts of within ±1-2 % dec$^{-1}$ in the northern mid- to high latitudes, and the southern high latitudes, and a positive but insignificant drift of up to 5 % dec$^{-1}$ in the tropics and southern mid-latitudes with overall uncertainties in this region ranging up to 5-10 % dec$^{-1}$ (2σ level) in the low stratosphere.

3.  Line 121, "… and concluded that v4.1/4.2 showed slightly larger biases than v3.5/3.6 in the middle to upper stratosphere". The ACE-FTS v4.1 and v3.6 ozone show the largest differences in

the middle stratosphere (~30km), with positive biases of ~3% and 9% for v3.6 and v4.1 ozone, respectively (see Figure 2 in Sheese et al (2022)). The v4.1 ozone is not slightly larger (i.e. 6% larger) than v3.6 in the middle stratosphere. The above sentence needs to be revised or indicates up to ~6% differences between v4.1 and v3.6 ozone in the middle stratosphere.

Rewording has been done for this line.

Original: "… concluded that v4.1/4.2 showed slightly larger biases than v3.5/3.6 in the middle to upper stratosphere, but is …"

Reworded: "… concluded that v4.1/4.2 showed larger biases than v3.5/3.6 by 2 – 6% between ~22 and 42 km, but is …"

4. Line 162, There is a typo in the Gaussian function, in the denominator square root of (2*pi*s) should be square root of (2*pi) *s  (i.e. "s" is outside of the square root). Another question is why s is set to be 1. The s value should correspond to the vertical resolution of ACE-FTS vertical resolution. Since the vertical resolution of ACE-FTS is ~3 – 4 km, the "s" value would be larger than 1 (see equation (2) in Sheese et al, 2017 (https://doi.org/10.1016/j.jqsrt.2016.06.026)). Using smaller s value would result in less smoothing ozonesond profiles (e.g. not comparable to ACE-FTS vertical resolution) and create larger differences (or artifacts) when comparing ACE-FTS ozone against ozonesondes in the regions with strong vertical gradient.

Yes, the "s" should be outside of the square root term.  This has been corrected.

The vertical resolution of ACE-FTS is ~3-4 km, based on the instrument field-of-view. However, the vertical sampling varies from 1.5 to 6 km with altitude and beta angle. With measurements taken every 2s, the lower part of the ACE-FTS profiles can be oversampled and provide an increased "effective vertical resolution" (Hegglin et al., 2008, Boone et al., 2023).

The suggestion of using a larger s value (such as s=3 or s=2) has been tested. These test results show that the larger smoothing parameters result in increased discrepancies between the ACE-FTS and ozonesonde data on the 1-km retrieval grid, particularly in the tropics. Thus, we have chosen to maintain use of s=1 for this smoothing function in the analysis.

These results comparing using s=1 and s=2 are shown in the plots below.

References:

Boone, C.D., Bernath, P.F., and Lecours, M.: Version 5 retrievals for ACE-FTS and ACE-imagers, Journal of Quantitative Spectroscopy and Radiative Transfer, 310, 108749, https://doi.org/10.1016/j.jqsrt.2023.108749, 2023.

Hegglin, M. I., Boone, C. D., Manney, G. L., Shepherd, T. G., Walker, K. A., Bernath, P. F., Daffer, W. H., Hoor, P., and Schiller, C.: Validation of ACE-FTS satellite data in the upper troposphere/lower stratosphere (UTLS) using non-coincident measurements, Atmos. Chem. Phys., 8, 1483–1499, https://doi.org/10.5194/acp-8-1483-2008, 2008.

[Figure]

*Figure 1: Version of Figure 6 from paper showing results for coincident comparisons using s=1 in the Gaussian function for smoothing the ozonesonde profiles.*

[Figure]

*Figure 2: Recalculation of Figure 1 (above) showing results using s=2 in the Gaussian function for ozonesonde profile smoothing. Note, the differences are greater than those shown in Figure 1.*

[Figure]

Figure 3: Figure 7 from paper showing aggregated results for coincident comparisons from Figure 1 (s=1).

[Figure]

Figure 4: Recalculation of Figure 3 (above) showing aggregated results from Figure 2 (s=2).

5. Line 199, "Laeng et al. (2021)". The published date is 2022 (i.e. Laeng et al. (2022)). Please double check the doi number of this paper (reference section in line 822). The doi number is "https://doi.org/10.5194/amt-15-2407-2022"

Corrected.

6. Line 454, "This is an interesting feature showing the differences appear to depend on the atmospheric temperature profiles". Ozone in the UTLS is mainly controlled by dynamics. The larger differences between ACE-FTS and ozonesondes in the UTLS (or tropopause height) could result from larger dynamic variability and sampling biases such as using more relaxed coincident criteria. This can be seen in Figure 7, where differences between ACE-FTS and ozonesondes in the UTLS (and tropopause) are smaller than those in Figure 9 due to stringent coincident criteria. The ACE-FTS ozone retrieval in the UTLS regions could also be affected by aerosol or cloud. The above sentence (line 454) needs more clarification.

This speculative sentence has been deleted.

Replace with: "Similar features, albeit with smaller magnitudes, are seen in the coincident profile comparisons in Figs. 7a-7e and may suggest that these differences result from dynamic variability in this region and the range of coincidence criteria used in this study."

7. Lines 601, "the aggregated mean instrument drifts show small insignificant drifts within ±1% dec$^{-1}$". The altitude regions for ±1% dec$^{-1}$ drift needs to be specified. For example a few kilometer above the tropopause.

This has been clarified by adding a few words about the northern mid-latitude and polar regions.

Revised: "…the aggregated mean instrument drifts show small insignificant drifts within ±1-2% dec$^{-1}$ from ~2 km above the tropopause up to ~30 km."